# Exploiting Gangliosides for the Therapy of Ewing’s Sarcoma and H3K27M-Mutant Diffuse Midline Glioma

**DOI:** 10.3390/cancers13030520

**Published:** 2021-01-29

**Authors:** Arthur Wingerter, Khalifa El Malki, Roger Sandhoff, Larissa Seidmann, Daniel-Christoph Wagner, Nadine Lehmann, Nadine Vewinger, Katrin B. M. Frauenknecht, Clemens J. Sommer, Frank Traub, Thomas Kindler, Alexandra Russo, Henrike Otto, André Lollert, Gundula Staatz, Lea Roth, Claudia Paret, Jörg Faber

**Affiliations:** 1Department of Pediatric Hematology/Oncology, Center for Pediatric and Adolescent Medicine, University Medical Center of the Johannes Gutenberg-University Mainz, 55131 Mainz, Germany; Arthur.Wingerter@unimedizin-mainz.de (A.W.); khalifa.elmalki@unimedizin-mainz.de (K.E.M.); Nadine.Lehmann@unimedizin-mainz.de (N.L.); Nadine.Vewinger@unimedizin-mainz.de (N.V.); Alexandra.Russo@unimedizin-mainz.de (A.R.); Henrike.Otto@unimedizin-mainz.de (H.O.); Lea.Roth@unimedizin-mainz.de (L.R.); faber@uni-mainz.de (J.F.); 2University Cancer Center (UCT), University Medical Center of the Johannes Gutenberg-University Mainz, 55131 Mainz, Germany; Thomas.Kindler@unimedizin-mainz.de; 3Lipid Pathobiochemistry, German Cancer Research Center, 69120 Heidelberg, Germany; r.sandhoff@dkfz.de; 4Institute of Pathology, University Medical Center of the Johannes Gutenberg-University Mainz, 55131 Mainz, Germany; Larissa.Seidmann@unimedizin-mainz.de (L.S.); Daniel-Christoph.Wagner@unimedizin-mainz.de (D.-C.W.); 5Institute of Neuropathology, University Medical Center of the Johannes Gutenberg-University Mainz, 55131 Mainz, Germany; katrin.Frauenknecht@unimedizin-mainz.de (K.B.M.F.); Clemens.Sommer@unimedizin-mainz.de (C.J.S.); 6Centre for Soft Tissue Sarcoma, GIST and Bone Tumors, Eberhard-Karls-University Tuebingen, 72076 Tuebingen, Germany; Frank.Traub@unimedizin-mainz.de; 7Department of Orthopaedic Surgery, Eberhard-Karls-University Tuebingen, 72076 Tuebingen, Germany; 8Center for Orthopedic and Trauma Surgery, University Medical Center of the Johannes Gutenberg-University Mainz, 55131 Mainz, Germany; 9German Cancer Consortium (DKTK), Site Frankfurt/Mainz, Germany, German Cancer Research Center (DKFZ), 69120 Heidelberg, Germany; 10Section of Pediatric Radiology, Department of Diagnostic and Interventional Radiology, University Medical Center of the Johannes Gutenberg-University Mainz, 55131 Mainz, Germany; andre.lollert@unimedizin-mainz.de (A.L.); gundula.staatz@unimedizin-mainz.de (G.S.)

**Keywords:** ganglioside, GD2, dinutuximab, eliglustat, miglustat, H3K27M-mutant diffuse midline glioma, Ewing’s sarcoma, osteosarcoma

## Abstract

**Simple Summary:**

Osteosarcoma, Ewing’s sarcoma, and H3K27M-mutant diffuse midline glioma are rare but aggressive malignancies occurring mainly in children. Due to their rareness and often fatal course, drug development is challenging. Here, we repurposed the existing drugs dinutuximab and eliglustat and investigated their potential to directly target or indirectly modulate the tumor cell-specific ganglioside GD2. Our data suggest that targeting and/or modulating tumor cell-specific GD2 may offer a new therapeutic strategy for the above mentioned tumor entities.

**Abstract:**

The ganglioside GD2 is an important target in childhood cancer. Nevertheless, the only therapy targeting GD2 that is approved to date is the monoclonal antibody dinutuximab, which is used in the therapy of neuroblastoma. The relevance of GD2 as a target in other tumor entities remains to be elucidated. Here, we analyzed the expression of GD2 in different pediatric tumor entities by flow cytometry and tested two approaches for targeting GD2. H3K27M-mutant diffuse midline glioma (H3K27M-mutant DMG) samples showed the highest expression of GD2 with all cells strongly positive for the antigen. Ewing’s sarcoma (ES) samples also showed high expression, but displayed intra- and intertumor heterogeneity. Osteosarcoma had low to intermediate expression with a high percentage of GD2-negative cells. Dinutuximab beta in combination with irinotecan and temozolomide was used to treat a five-year-old girl with refractory ES. Disease control lasted over 12 months until a single partially GD2-negative intracranial metastasis was detected. In order to target GD2 in H3K27M-mutant DMG, we blocked ganglioside synthesis via eliglustat, since dinutuximab cannot cross the blood–brain barrier. Eliglustat is an inhibitor of glucosylceramide synthase, and it is used for treating children with Gaucher’s disease. Eliglustat completely inhibited the proliferation of primary H3K27M-mutant DMG cells in vitro. In summary, our data provide evidence that dinutuximab might be effective in tumors with high GD2 expression. Moreover, disrupting the ganglioside metabolism in H3K27M-mutant DMG could open up a new therapeutic option for this highly fatal cancer.

## 1. Introduction

Gangliosides, such as GD2, are glycosphingolipids composed of ceramide and an oligosaccharide complex containing sialic acids like N-acetyl neuraminic acid and are found in the outer leaflet of plasma membrane (Figure 1). Almost 200 gangliosides species have been described, showing differences in the number, the order, and the linkage of the glycosyl and sialyl residues [1].

GD2 has limited expression in normal tissues but is overexpressed across a wide range of tumors particularly of neuroectodermal origins such as neuroblastoma melanoma and small-cell lung cancer [2,3]. Due to its restricted expression in normal tissue, GD2 has long been recognized as an important target for cancer immunotherapy [4]. The use of the chimeric monoclonal anti-GD2 antibodies ch14.18/SP2/0 (dinutuximab) and ch14.18/CHO (dinutuximab beta) is considered to be standard of care in the first-line treatment of children with high-risk neuroblastoma [5,6]. Additionally, humanized anti-GD2 antibodies and anti-GD2 CAR-T cells are in clinical testing (NCT03363373, NCT02107963, NCT02502786). Dinutuximab shows activity in combination with chemokines and with temozolomide (TMZ) + irinotecan (IRN) [7,8]. Recently, the FDA granted accelerated approval to naxitamab (humanized murine IgG3 antibody m3F8) for high-risk neuroblastoma in bone or bone marrow based on response rate results in two trials (NCT03363373 and NCT01757626). 

In the pediatric population, GD2 has been discussed as target in rare and aggressive malignancies such as osteosarcoma (OS), Ewing’s sarcoma (ES), and H3K27M-mutant diffuse midline glioma (H3K27M-mutant DMG). ES represents the second common malignant bone tumor in children and young adults, being associated with a poor outcome especially in patients with metastatic disease and even poorer in relapsed patients despite aggressive multimodal treatment regimens including surgery, chemotherapy, and radiation [9]. In ES, GD2 has been suggested as target antigen to eradicate micrometastatic cells and prevent relapse in high-risk disease [10]. However, anti-GD2 antibody treatment is hardly used [11]. A phase I clinical trial (NCT00743496) from the St. Jude Clinical Research Hospital investigating the anti GD 2 antibody hu14.18K322A and aiming to enroll also ES patients was completed in 2014. However, recently published data considered neuroblastoma and osteosarcoma patients only [12]. Especially for H3K27M-mutant DMG, a rare malignant brain tumor in children and adults, novel treatment strategies are urgently needed. Radiotherapy is considered to be standard of care but provides even in combination with various systemic antineoplastic agents unsatisfactory treatment results with a median overall survival of around 12 months [13]. Recently, GD2 has been discussed as a target for CAR-T cell therapy in H3K27M-mutant DMG [14]. In OS, GD2 has been suggested to play a role in chemotherapy resistance and tumor progression [15]. Phase I clinical trials conducted with melanoma and osteosarcoma patients showed response to anti-GD2 antibody treatment, but only in a fraction of the treated patients [16,17]. This indicates the significance of a proper stratification to identify eligible patients for the anti-GD2 treatment. 

The anti-GD2 antibodies work by inducing antibody-dependent cellular cytotoxicity (ADCC) and complement-dependent cytotoxicity (CDC) [18,19], and activity of dinutuximab or naxitamab is expected in tumors with high density of GD2 expression, independently of the function of GD2. Therefore, patients with high and homogeneous GD2 expression on tumor tissues may respond to treatment with dinutuximab. This would not apply to GD2-positive brain tumor patients, since anti-GD2 antibodies do not cross the blood–brain barrier [20]. To overcome this limitation, application of antibodies either into the cerebrospinal compartment in case of leptomeningeal disease or directly through the interstitial spaces of the CNS into parenchymal masses are reasonable approaches, having been subject of early clinical trials [21,22,23]. 

Beyond antibody-based approaches, a novel therapeutic concept might be the inhibition of the pathway leading to GD2 and more generally to gangliosides synthesis (Figure 1C). Many studies indicate that tumor-associated gangliosides are a result of initial oncogenic transformation and play a key role in tumor progression [24]. Accordingly, inhibition of GD2 and GD3 synthesis has been suggested as a therapeutic approach in several cancers [25,26]. However, methods to inhibit gangliosides in a clinical setting in the context of cancer are largely missing. One of the first steps in the metabolic pathway of gangliosides is the conversion of ceramide to glucosylceramide [1]. Thus, inhibition of the glucosylceramide synthase could be lethal in cells with high GD2 expression. Small molecule inhibitors of glucosylceramide synthase are currently used for treating children with the lysosomal storage disorders Niemann Pick’s and Gaucher’s disease [27].

Here we analyzed the expression and druggability of GD2 in different pediatric tumor entities. Our data indicate that ES patients could benefit from treatment with dinutuximab, while H3K27M-mutant DMG patients might benefit from treatment with clinically available glucosylceramide synthase inhibitors. 

## 2. Results

### 2.1. Validation of an Anti-GD2 Antibody

The monoclonal antibody clone 14.G2a used for flow cytometric detection of GD2 in this work was commercially available (see materials and methods). A murine anti-GD2 MoAb 14.18 was developed against the human neuroblastoma cell line LAN-1, and 14.G2a, the IgG2a class switch variant of 14.18, was designed with the goal of mediating enhanced ADCC [28]. The antibody was validated on lipids separated by thin layer chromatography (Figure 2). Among the tested gangliosides, the antibody recognized only GD2. However, the GD2 standard appears to contain an antibody-positive band, which is not stained with orcinol. Since the GD2 standard very likely was purified from a biological source, it may contain at low abundance a very similar structure. This band is migrating at the height of GM1, but is not stained with orcinol. Since GM1 in lane 1 is not recognized by the antibody, it must be a different structure. GD3, migrating under these conditions between GM1 and GD1a, was tested in a different experiment (Appendix A) and can also be excluded. Eventually it could be acetylated GD2, which would migrate faster than GD2 itself.

### 2.2. Ewing’s Sarcoma and H3K27M-Mutant DMG Highly Express GD2

The antibody described in 2.1 was used for detection of GD2 by flow cytometry on primary cell lines. We analyzed seven osteosarcoma (OS), 6 ES, one central nervous system high-grade neuroepithelial tumor with BCOR alteration (HGNET-BCOR), one dysembryoplastic neuroepithelial tumor (DNET), and two H3K27M-mutant DMG (Table 1). The percentage of cells expressing GD2 is shown in Table 1. The gating strategy is shown in Appendix A. H3K27M-mutant DMG carry a substitution of methionine for lysine at site 27 of histone 3 (H3K27M). Cultured cells from two H3K27M-mutant DMG and DNET were validated by immunohistochemistry (IHC), and immunoreactivity (IR) was compared to the patients’ previous diagnostic biopsy. IHC of H3K27M-mutant DMG cells of both patients revealed glial origin (GFAP), high proliferative activity (Ki67), as well as the presence of H3K27M mutant protein (Figure 3A). DNET cells showed expression of glial (GFAP) and—to a lesser extent—also neuronal markers (Neurofilament and Synaptophysin) (Appendix A), indicating that not only the glial but also a minor neuronal cell component was maintained in the cell culture. HGNET-BCOR cells were validated as described in [29]. Both H3K27M-mutant DMG samples had a high expression of GD2 with all cells being positive for the antigen (Figure 3B). A HGNET-BCOR sample and a DNET sample were negative and highly positive, respectively (Figure 3B). Difference in the expression in central nervous system (CNS)-tumors may mirror a different origin of the tumor. DNET are glioneuronal tumors, H3K27M-mutant DMG possibly derive from oligodendroglial lineage [30], and both have a neuroectoderm origin. In contrast, CNS-HGNET-BCOR are generally described as mesenchymal tumors. ES cells were validated by morphological and immunocytochemical analysis and were strongly positive for CD99 (Figure 4A), which is used in routine diagnostic to distinguish the Ewing family of tumors from other tumors of similar histological appearance [31]. Some ES samples showed high GD2 expression, but not all tumor cells were positive in all patients (Figure 4B). Osteosarcoma cells were validated as described in [32]. Osteosarcoma had low to intermediate expression with a high percentage of cells lacking the antigen (Figure 5). One way ANOVA and Tukey’s multiple comparison test found a statistically significant difference only between OS and H3K27M-mutant DMG (*p* = 0.00384), due to the low size of the groups. Expression of GD2 in commercially available cell lines is shown in Appendix A. The ES cell line did not express GD2. One of the two OS cell lines had a high but heterogeneous GD2 expression. The glioblastoma cell line U87-MG had an intermediate GD2 expression.

In conclusion, these data indicate that although GD2 is expressed in several pediatric tumors with a poor outcome, heterogeneity and level of expression is patient-specific. H3K27M-mutant DMG and some ES show a uniform and high GD2 expression and are candidates for therapies addressing GD2.

### 2.3. Anti-GD2 Antibody Treatment in a Patient with Ewing´s Sarcoma

A five year old girl with a primary refractory ES of the first left rib (Appendix A) who had developed very early local progressive disease during consolidation treatment with the Ewing 2008 treatment protocol underwent an individualized salvage therapy regimen at our pediatric oncology department including off-label use of dinutuximab beta (for detailed information see “Material and methods, Section 4.9”). Symptomatic progressive disease was initially addressed by surgery (incomplete resection of the intraspinal part of the tumor) immediately followed by two cycles of irinotecan/temozolomide (IT) and irradiation of the primary tumor site including the entire left hemithorax. 

Since primary refractory disease in ES is associated with extremely poor outcome and no standardized curative salvage treatment is available [33], an individualized treatment concept was evaluated for the patient. Molecular analysis of the most recently resected tumor tissue did not elucidate any druggable targets in the framework of the INFORM platform [34]. 

However, a high level of GD2 expression that was identified by flow cytometry analysis of primary tumor cells pointed towards GD2 antibodies as a promising treatment option for this patient (sample no. 408, Figure 4B). Since irinotecan and temozolomide were considered to be effective in relapsed ES [35,36,37,38] on the one hand and to be safe and effective in combination with dinutuximab in children with relapsed and refractory neuroblastoma on the other hand [7], an individualized treatment protocol was designed including irinotecan, temozolomide, and dinutuximab beta. The aim of the treatment was to reach stable disease at least and concomitantly provide the best possible quality of life, minimizing hospitalization and acute side effects. Combination of irinotecan/temozolomide and dinutuximab beta was introduced approximately 12 weeks after surgery and four weeks after completion of irradiation (Figure 6A). The first cycle was conducted in an inpatient setting due to expected acute side effects of pain, fever, and hypersensitive reaction. Despite routine supportive medication, the patient presented with fever and mild oxygen desaturation with the lowest blood oxygen level of 89%, requiring oxygen therapy for a few days. Although computer tomography (CT) of the chest revealed signs for radiation-induced pneumonitis, the patient stayed clinically stable and symptoms resolved rapidly. As supportive standard of care, pain was adequately controlled with gabapentin, acetaminophen, and morphine. In the later cycles, morphine was no longer necessary. Subsequent treatment could be safely carried out in an outpatient setting. The Lansky performance status score sustainably improved to a level of 80% during the treatment. Except for Horner´s syndrome, which had appeared after the very first surgery, the patient did not exhibit neurological symptoms at any time. 

To assess treatment efficacy of the personalized protocol, magnetic resonance imaging (MRI) of the chest and thorax was performed after every second cycle. Additionally, a positron emission tomography (PET) scan was carried out. Remaining tumor volume decreased after radiation therapy and stayed stable over the next 10 month (Figure 6B–D). No active tumor could be detected by PET scan in week 26 after surgery. In total, the patient’s progression-free survival (PFS) was approximately 12 months, until a single metastasis in the right parietal skull was detected by a routine whole-body MRI (Figure 6E). Only four weeks later, after being off systemic treatment for eight weeks, multiple bone metastases were detected (Appendix A).

### 2.4. GD2 Expression and Immune Cells Infiltration in the Dinutuximab-Refractory Metastasis

To understand the mechanism of resistance associated with the development of the skull metastasis under the therapy with dinutuximab, we analyzed the expression of GD2, the composition of glycolipids and the profile of infiltrating immune cells before and after the treatment with dinutuximab. Two tumor samples before the dinutuximab therapy were available for analysis, one from a relapse in the spine (no 408T, used for the isolation of primary tumor cells no 408) and one from a relapse in the rib (no 400T, no primary cells available). Concerning the intracranial metastasis, the tumor itself (no 482T) and primary tumor cells established from it (no 482) were available for analysis. Flow cytometric analysis of GD2 expression indicated a homogeneous and strong GD2 expression in the primary cells isolated from the tumor before the administration of dinutuximab (Figure 7A, sample no 408) and the presence of a cell population with a reduced GD2 expression in the primary cells isolated from the intracranial metastasis (Figure 7A, sample no 482). Importantly, all primary cells of sample no 482 were validated as ES cells, suggesting that the presence of a population with reduced GD2 expression is not due to a contamination of the primary cells with a non-relevant cell population (Figure 4A). The EWSR1-FLI1 fusion was detected in both samples, confirming the ES diagnosis [31] (Appendix A). Mass spectrometric analysis of glycolipids indicated that the metastasis had increased GM3 levels but otherwise decreased levels of more complex gangliosides (Figure 7B). 

We analyzed the immune cell infiltrates in longitudinal samples (400T versus 482T), and distinguished between tumor and non-tumor (i.e., stroma and necrosis, Appendix A). Immune cells infiltration was heterogeneous within the same tumor, but we observed a tendency to decreased numbers of B cells, NK cells, and macrophages in the sample that was resected after dinutuximab treatment (Figure 8). By contrast, the overall neutrophil density (cells per square millimeter) was higher, but the number of tumor-associated neutrophils was lower after dinutuximab therapy. The amount of CD163-positive (“M2”) macrophages increased both in the tumor and non-tumor compartment following dinutuximab, whereas the density of cytotoxic T cells remained stable at a low level (Figure 8).

### 2.5. Perturbation of the Glycosphingolipid Metabolism Inhibits the Growth of H3K27M-Mutant DMG

Our analysis indicates a strong and homogeneous expression of GD2 in H3K27M-mutant DMG. However, dinutuximab does not cross the blood–brain barrier and cannot be used to treat patients with CNS involvement. Due to the very high expression of GD2, we speculated that the glycosphingolipid metabolism is particularly active in these cells and that its perturbation might be lethal for the cells. We therefore inhibited the glycoysphingolipids synthesis in vitro with eliglustat (Cerdelga^®^), an inhibitor of the ceramide synthase (IC50 = 24 nM) that is used for the treatment of pediatric patients with Niemann Pick’s and Gaucher’s disease [27]. For details on gangliosides metabolism, see Figure 1A,C. Eliglustat acts as a substrate reduction therapy by reducing the production of glucosylceramide (Figure 1C). Eliglustat completely inhibited cell growth at a concentration of about 45–61.5 µM (IC50) as demonstrated by impedance analysis (Figure 9), while cell growth at lower concentration (≤10 µM) was still observed.

## 3. Discussion

In 2009, the U.S. National Cancer Institute ranked GD2 at position twelve among 75 potential targets for anti-cancer therapy based for example on potential therapeutic effect, specificity, degree of expression, and immunogenicity [39]. So far, GD2 can be targeted only by immunotherapy, and no other targeted therapy exists. Moreover, GD2 antibodies have gained regulatory approval exclusively for neuroblastoma so far. Our work may help to prioritize further pediatric tumor entities for cancer immunotherapy with anti-GD2 antibodies and discloses a new approach for regulating gangliosides in cancer using small molecules inhibitors.

*Validation of an anti-GD2 antibody*. It cannot be completely excluded that the anti-GD2 antibody may cross react with structurally similar glycans or other mimicries. Among the gangliosides tested in this work, the antibody recognized only GD2, confirming and expanding previous data [40]. In particular, it did not recognize GM2 or GD1b, which are either missing one of the two sialic acids or containing an additional galactosyl residue, respectively. Our data indicate that the antibody may recognize acetylated GD2, which in contrast to GD2 is not detected on peripheral nerves [41]. However, further experiments are needed to confirm this hypothesis.

*Ewing’s Sarcoma and H3K27M-mutant DMG highly express GD2*. Expression of GD2 in several cancer entities has been reported; however, the results are not always consistent, probably due to the methods used to detect GD2. Indeed, gangliosides are soluble in some solvents used to fix tissues, and inconsistencies in the IHC-measured GD2 levels have been described [42]. For example, in the case of osteosarcoma, some studies indicate higher GD2 expression in recurrent osteosarcomas compared with initial biopsies while others do not [15,43]. Accordantly, a heterogeneous response has been observed in OS patients treated with a humanized anti-GD2 antibody [12]. Flow cytometry on living cells, as performed in our study, overcomes the limitations in GD2 detection caused by sample processing. Detection of GD2 by IHC on frozen tumor samples could also represent a valid alternative, as the tumor is not manipulated before GD2 analysis [42]. Our data show that the heterogeneity and level of GD2 expression in ES and OS is patient- rather than entity- specific. GD2 is considered a differentiation-related antigen during neuronal maturation so that GD2 expression likely indicates the presence of less differentiated cells derived from neuronal crest. Accordingly, GD2 expression in neuroblastoma depends on the differentiation status, with the strongest staining observed in undifferentiated neuroblastoma [42]. In OS, primitive cells are positive, whereas more mature foci do not express GD2 [42]. Thus, the degree of differentiation of the tumor cells may explain the heterogeneous expression of GD2 within the same tumor. This heterogeneity likely affects the effectiveness of a GD2-based immunotherapy, as already shown in neuroblastoma, where sensitivity to natural killer cells (NK)-mediated lysis is dependent on the proportion of GD2-positive cells [44]. Interestingly, in ES and H3K27M-mutant DMG, GD2 expression has been linked to aberrant histone methylation. In ES, Enhancer of zeste homolog 2 (EZH2), a histone-lysine N-methyltransferase, is a negative regulator of GD2 [45]. High-level EZH2 expression is induced in ES as a direct consequence of the EWSR1-FLI1 fusion [46]. In diffuse midline gliomas, GD2 expression is associated with the presence of the H3F3A K27M and the HIST1H3B K27M mutation, while H3 wild type samples barely express GD2 [14]. GD2 overexpression in K27M mutants is related to transcriptional perturbation resulting from the histone mutation that induces upregulation of ganglioside synthesis enzymes [14].

*Anti-GD2 antibody treatment in a patient with Ewing´s sarcoma*. We applied dinutuximab beta to a patient with refractory ES whose tumor had a high and homogeneous expression of GD2. Preclinical data suggest that ES can be targeted by anti-GD2 antibodies. The combination treatment with zoledronic acid, IL-2, and anti-GD2 antibody ch14.18/ CHO was shown to exert Vδ2+ T cells dependent ADCC against GD2-expressing ES cell lines. The cytotoxic effect correlates with the intensity of GD2 expression. In a xenografts mouse model, GD2-expressing ES combination of adoptively transferred Vδ2+ T cells expanded in vitro with zoledronic acid and IL-2, with anti-GD2 antibody ch14.18/ CHO, and with systemic zoledronic acid, significantly suppressed tumor growth [47]. Additionally, there is strong evidence that various cytotoxic agents are able to stimulate anticancer immune responses involving NK cells and CD8 positive T cells and thus to augment immunotherapeutic effects [48,49]. We achieved disease control over a period of about 12 months. Historical data from small cohort analyses indicated a median time to progression of patients with relapsed or refractory ES who had received IT for treatment, ranging between 4.6 and 8.3 months [35,36,38,50]. A retrospective analysis of 51 relapsed ES patients calculated a PFS of 7.7 months for the pediatric subgroup [37]. However, interim analyses from an international randomized controlled trial of chemotherapy for the treatment of recurrent and primary refractory Ewing’s sarcoma (rEECur) revealed a median PFS of 4.7 months (95% CI: 3.4 to 5.7) for patients who had been treated in the IT arm [51]. Given this poor outcome, achieved treatment result in our patient by incorporating dinutuximab beta in an individualized treatment approach is quite remarkable, indicating a synergistic antineoplastic effect of IT and anti-GD2 antibodies in GD2 expressing ES. Noteworthily, two additional metastatic bone lesions were newly revealed within a very short time interval of four weeks, only eight weeks after systemic treatment had been stopped. This suggests that combination treatment with IT and dinutuximab beta likely still exhibited, even though unsatisfactorily, antitumoral activity during progression of disease. Importantly, treatment could be safely applied predominantly in an outpatient setting and thus significantly improved the patient’s quality of life. This experience further supports the need for a prospective clinical evaluation of anti-GD2 antibodies in combination with a sufficient conventional cytotoxic treatment in advanced ES, as recently recommended by the COG task force [52].

Monoclonal antibodies mediate tumor cell killing through ADCC, which is mainly induced by FcγR-expressing NK cells or macrophages. Dinutuximab beta is licensed in Europe for first-line treatment in patients with high-risk neuroblastoma, a GD2-positive malignant pediatric cancer, with and without a combination of interleukin 2 (IL-2). IL-2 increases the number and in vitro tumoricidal activity of NK cells and in vitro ADCC mediated by peripheral blood mononuclear cells obtained from patients with cancer can be enhanced dramatically if the patients were treated with IL-2 [53]. We combined dinutuximab beta with IT while not including IL-2 to minimize the risk and intensity of acute side effects like fever, capillary leakage syndrome, pain, and hypotension. Whether the addition of IL-2 or granulocyte-macrophage-colony stimulating factor (GM-CSF), which also increase monocyte cytotoxicity [54], may improve the efficacy of anti-GD2 antibodies in ES remains to be elucidated.

### GD2 Expression and Immune Cells Infiltration in the Dinutuximab-Refractory Metastasis

The therapeutic activity of monoclonal antibodies is limited by evolving resistance mechanisms. These mechanisms include loss of expression of the antigen and impaired NK cell-mediated antitumor response. So far, there are no preclinical or clinical data defining the level of GD2 expression required to trigger antitumor responses. The selective pressure of dinutuximab therapy may result in decreased GD2 expression, as observed with targeting CD19 with CAR T cells in leukemia [55]. In neuroblastoma, a low percentage of GD2-positive cells prior to immunotherapy was associated with relapse in patients receiving anti-GD2 immunotherapy [44]. In the case described in this study, the cranial metastasis that developed under dinutuximab beta expressed less GD2 compared to the primary tumor, with about 40% of the tumor cells showing reduced GD2 expression. As we observed an increase in GM3 expression in the metastasis, reduction in GD2 expression might be associated with a down regulation of *ST8SIA1* and *B4GALNT1,* which are required to transform available GD3 into GT3 or GM2, respectively (Figure 1C). This hypothesis, however, needs to be further validated. Recently, a two-gene signature composed of *ST8SIA1* + *B4GALNT1* has been suggested as efficient predictor of GD2-positive phenotype [56]. It is possible that GD2-negative clones formed the metastasis in the first instance, and that GD2 was subsequently re-expressed due to an uneven distribution of large molecules such as antibodies in huge tumors. Indeed, resistance to antibody-targeted cancer therapies has been linked to a non-homogeneous distribution of the antibody inside the tumor, resulting in an untargeted subpopulation of cancer cells [57,58]. 

An ineffective infiltration of immune cells in bulky and necrotic tumor mass may also explain an ineffective antitumor response. The cellular effectors potentially exerting ADCC includes CD3−CD56+ NK lymphocytes, CD3+CD16+ T-cell subset, CD16+CD33+ macrophages, and CD16+ granulocytes. Macrophages can induce antibody-dependent cellular phagocytosis (ADCP) and engage the adaptive arm of the immune system, and M1 macrophage infiltration has been shown to support the cytotoxic activity of trastuzumab in responsive tumors [59]. NK and T cells expressing CD16 constitute the effector lymphocytes mainly involved in short-term trastuzumab-dependent cytotoxicity [60]. In the case presented here, we observed decreased densities of immune cells that contribute to antibody-mediated anti-tumor effects after treatment with dinutuximab, including NK cells, tumor-associated neutrophils (TAN), and macrophages (TAM). The reallocation of neutrophils in post-treatment specimen with increased numbers in stromal and necrotic areas might also suggest reduced ADCC activity, since the destruction of opsonized cells requires a direct contact between tumor cells and neutrophils [61]. On the other hand, the immunohistochemical markers used in this study cannot differentiate between anti- and pro-tumorigenic myeloid cells, including myeloid-derived suppressor cells (MDSC) and certain TAN and TAM subsets that promote tumor growth and metastasis amongst others by suppression of adaptive immune functions and extracellular matrix remodeling [62]. We found only low densities of M2-like macrophages, but their slight increase following antibody treatment might be an indication for effective ADCP, since ongoing phagocytosis alters macrophage function and phenotype into this direction [63]. It has been described that the therapeutic effects of monoclonal antibodies also depend on the induction of adaptive immune responses, including the presence of cytotoxic T cells in the tumor [64]. In our case, very low infiltration of T and B cells was observed, indicating an attenuated adaptive anti-tumor immunity independent of antibody treatment. Finally, we cannot judge whether the observed changes of tumor-associated immune cells are a consequence of successful antibody-mediated anti-tumor effects or simply an epiphenomenon of an altered site-specific microenvironment in the metastasis. 

*Perturbation of the glycosphingolipid metabolism inhibits the growth of H3K27M-mutant DMG.* H3K27M-mutant DMG samples had a strong and homogeneous expression of GD2, which has already been described [14]. Monoclonal antibodies do not efficiently cross the blood–brain barrier, and therefore dinutuximab is not suitable for treating H3K27M-mutant DMG or other CNS tumors. Activated T cells can infiltrate the CNS following adoptive transfer and CAR-T cells targeting GD2 are in development [65]. However, preclinical studies in mice have raised concerns about the toxicity of GD2-targeting CAR T-cells in H3K27M-mutant DMG due to tumor-associated inflammation generated by CAR T-cell activity [14]. Moreover, the expression of GD2 in a normal brain has been associated to an extensive CAR T-cells infiltration and proliferation within the brain, leading to neuronal destruction in preclinical models of neuroblastoma [66]. Even if the neurotoxicity of GD2 CARs is still a matter of debate [67,68], other approaches are needed to address GD2 in CNS tumors.

Gangliosides play a role in cancer development by regulating angiogenesis, cell adhesion/motility, and signal transduction [24]. For example, gangliosides have been described as key drivers for glioblastoma stem cells and tumorigenicity [69]. Induction of ganglioside GM2/GD2 enhances tumor incidence and growth speed of melanoma in vivo [70]. Therefore, enzymes of the ganglioside synthesis could represent realistic targets for the development of antitumoral reagents. One of the first steps in the metabolic pathway of gangliosides is the conversion of ceramide to glucosylceramide. Here, we hypothesized that due to the extremely high expression of GD2 in H3K27M-mutant DMG, the glycosphingolipids metabolism is particularly active in these cells and that its perturbation via inhibition of the glucosylceramide synthase could be lethal. The sphingolipid oncometabolism, which has recently been described as a metabolic vulnerability in cancer and tumor growth, was inhibited by eliglustat in an in vitro and a preclinical in vivo model of prostate cancer [71]. In line with our data (IC50 45–61.5 µM), the cytotoxic concentration of eliglustat required to inhibit the prostatic cancer cell line RM-9 in vitro was 128 µM. Despite the high amount of eliglustat needed to induce this in vitro cytotoxicity, RM-9 tumor growth in vivo was still suppressed by using eliglustat at a therapeutic dosage, and tumor tissues showed reductions in glycosphingolipids. Of note, glucosylceramide synthase inhibitors are already used in the clinic. Eliglustat and miglustat are indeed approved for the treatment of pediatric patients with Gaucher’s disease, the most common lipid storage disorder resulting from a genetic deficiency of the enzyme glucocerebrosidase (glucosylceramidase) [72]. In the context of Gaucher’s disease, eliglustat and miglustat are used in the so-called substrate reduction therapy, based on partial reduction of the synthesis of glucosylceramide and hence of subsequent metabolites [73] (Figure 1C). Only miglustat achieves significant distribution into the brain [74]. Further CNS-penetrant glucosylceramide synthase inhibitors are currently in development but have not been approved so far [75]. Targeting the glycosphingolipids metabolism in H3K27M-mutant DMG with CNS-penetrant glucosylceramide synthase inhibitors may accelerate the access of H3K27M-mutant DMG patients to new therapies for this highly aggressive tumor. In the context of lipid storage disorder diseases, the treatment with glucosylceramide synthase inhibitors is generally well tolerated, with major toxicities related to the gastrointestinal tract that can be managed with a particular diet. Whether the treatment has side effects in patients without underlying enzymatic dysfunction needs to be clarified in further clinical trials. This could be important, because gangliosides are needed for neural cell function. However, neurotoxicity under treatment with eliglustat was not described in C57BL/6N mice without enzymatic dysfunction [71].

Taken together, our work underlines the importance of GD2 as a target for biomarker-driven personalized therapy protocols and reveals a previously neglected connection between the sphingolipid metabolism and its regulation in CNS-tumors via drugs that are already in clinical use.

## 4. Materials and Methods

### 4.1. Patients and Material

This study was performed in agreement with the declaration of Helsinki on the use of human material for research. In accordance with the ethics committee of Rhineland-Palatinate, written informed consent of all patients or their custodians was obtained for “scientific use of tumor tissue not needed for histopathological diagnosis” in the admission contract of the University Medical Center Mainz (§ 14 AVB). Tumor tissues used in this study are 400T, 408T and 482T, isolated from the ES patient described in Section 4.8 before (400T, spine, 408T, rib) or after (482T, skull) dinutuximab therapy. For samples provided from the University of Tübingen, ethical approval was obtained by the local ethics committee (Nr. 008/2014BO2). Commercially available cell lines were provided by ATCC.

### 4.2. Isolation and Cultivation of Primary Tumor Cells

Isolation of primary tumor tissue was performed by a pathologist from fresh tumor material using the cryosections as control to identify the regions containing vital tumor cells. Primary tumor cells of H3K27M-mutant DMG were isolated from a needle biopsy. The cells were isolated by mechanical tissue dissociation with GentleMACS dissociator (Miltenyi Biotec, Bergisch-Gladbach, Germany). Tissue of CNS tumors was enzymatically dissociated with 0.25% Trypsin (Thermo Fisher Scientific, Waltham, MA, USA) and DNase I (Sigma-Aldrich, St. Louis, MO, USA), whereas tissue of other tumor entities was enzymatically dissociated with L Liberase (Roche Applied Science, Penzberg, Deutschland), HBSS medium (Thermo Fisher Scientific, Waltham, MA, USA), and DNase I. Isolated cells were cultured in Dulbecco’s Modified Eagle’s Medium (DMEM) with 10% human serum, 1% L-glutamine, and 1% Penicillin-Streptomycin (all Thermo Fisher Scientific). Cells were diluted 1:2 to 1:3 for passaging and maintained in a humidified incubator with 5% CO_2_ at 37 °C. Primary cultures were used until passage five. 

### 4.3. Flow Cytometric Analysis of GD2

In order to prepare the primary cells for GD2 analysis, the cells were cultured until 70–80% confluency. The cells were gently rinsed twice with PBS (“Dulbecco’s Phosphate Buffered Saline”, Sigma-Aldrich Co., MO, USA), detached with 0.05% Trypsin-EDTA (Gibco™, Thermo Fisher Scientific, MA, USA), and harvested in DMEM. After centrifugation, 2 × 10^5^ to 5 × 10^5^ cells were resuspended in DMEM medium and kept in a humidified incubator with 5% CO_2_ at 37 °C for 30 min before GD2 analysis. Subsequently, cells were centrifugated, resuspended in 100 µL PBS, and incubated with 5 µL Fc Block^TM^ (BD Biosciences, Franklin Lakes, NJ, USA) for 10 min at room temperature. Afterwards, cells were stained with 5 µL anti-GD2 antibody (BD Biosciences, NJ, USA) and 10 µL 7-AAD (Beckman Coulter, Brea, CA, USA). After one washing step, cells were analyzed on Navios flow cytometer (Beckman Coulter, CA, USA). Unstained cells of each sample were used as negative controls. Data was visualized as histograms with FlowJo^TM^ (BD Biosciences, NJ, USA), which were represented in percent normalized to mode. The percentage of GD2 positive cells was calculated as shown in Appendix A and was used for GD2 expression scoring, as indicated in Table 1.

### 4.4. Lipid Analysis

Ganglioside GD2 mixture was obtained from Cayman chemicals (Michigan 48108 USA), GM3 was obtained from Matreya (Pennsylvania 16803 USA), and GM2 had been isolated from human GM2 gangliosidosis brain. Bovine brain cortex gangliosides “Cronassial^®^” had been obtained formerly from the company Dr. Madaus and Co. (Cologne, Germany). Alkaline-phosphatase-conjugated affinipure goat anti mouse (IgG+IgM) antibody was purchased from Jackson ImmunoResearch Europe Ltd. (Cambridgeshire, CB7 4EX United Kingdom). SigmaFAST BCIP/NBT was obtained from Sigma-Aldrich (82024 Taufkirchen, Germany).

#### 4.4.1. Lipid Extraction

Tumor tissue of up to 150 mg wet weight was homogenized in polypropylene tubes together with 1200 µL ice cold distilled water and a stainless-steel bullet using TissueLyser instrument two times for 2 min at 25 Hz. An aliquot of 50 µL was removed to determine total protein concentration with a BCA-assay. An aliquot corresponding to 25 mg wet weight was transferred to a new polypropylene cup for lipid extraction. After freeze drying, tissue was suspended in 500 µL chloroform:methanol:water (10:10:1) and treated four times for 3 min with ultrasound in a water bath at 37 °C in a total time of 20 min. Samples were centrifuged in a tabletop centrifuge (Eppendorf type 5415 C) at maximum speed. Supernatant was collected and the pellet was re-extracted likewise with chloroform:methanol:water (10:10:1) and in a third round with chloroform:methanol:water (30:60:8). Pooled lipid extract was then separated into neutral and anionic lipids using a DEAE sephadex cartridge. Both the fraction of neutral and of anionic lipids were then desalted using C18-reversed phase cartridges. Desalted lipid fractions were dried with a gentle nitrogen stream and taken up in chloroform:methanol:water (10:95:5) for further analysis.

#### 4.4.2. Thin Layer Chromatography and Immune Overlay

Lipids were loaded on TLC using a CAMAG Linomat IV. TLC plates were then pre-run with chloroform:acetone (1:1) until the top, dried carefully and developed with the running solvent chloroform:methanol: 0.2% aqueous CaCl2 (45:45:10). Plates were dried and fixed with Plexigum 28, as described previously [76]. Dried plates were blocked with 1% BSA in PBS for 30 min and incubated with anti-GD2 antibody diluted 1:50 at 4 °C overnight. Plates were washed three times with PBS containing 0.05% Tween 20 and incubated with alkaline phosphatase-conjugated secondary antibody for 2 h, washed three times with PBS containing 0.05% Tween 20, and developed with SigmaFast BCIP/NBT reagent (one tablet dissolved in 10 mL distilled water). 

#### 4.4.3. HILIC-Coupled Tandem Mass Spectrometry of Gangliosides

Aliquots of the acidic lipid extracts corresponding to about 6 mg wet weight were suspended in 200 µL chloroform:methanol:water (10:10:1) and analyzed with a Aquity I-class ultra-performance LC and a Xevo TQ-S “triple-quadrupole” instrument, both from Waters, using a CSH C18 column (2.1 × 100 mm, 1.7 µm; Waters). Lipids were separated in HILIC mode on a CORTECS HILIC column (2.1 × 150 mm, 1.6 µm; Waters) at a flow rate of 0.4 mL/min using a gradient starting with 95% solvent A (0.003% acetic acid in Acetonitrile) and 5% solvent B (10 mM ammonium acetate and 0.003% acetic acid in water), which increased from 0.5 min to 12 min to 70% B. From 13 to 14 min, it decreased again to 95% A and kept this ratio until 25 min before injecting the next sample. Gangliosides were detected in negative electrospray mode by single reaction monitoring multiple SRM mode (MRM) using the transition of the deprotonated molecular ion (GM3, GM2, GM1) or the double deprotonoated molecular ion (GD3, GD2, GD1, GT1) to the dehydrated and deprotonated N-acetyl neuraminic acid fragment (m/z 290.1) and identified by retention time of corresponding standard gangliosides. For the transition of GM3, GM2, GM1, GD3, GD2, GD1, and GT1, a collision energy of 50, 55, 65, 30, 33, 35, and 32 eV was used. Peak areas were normalized to corresponding total protein content before relative comparison.

### 4.5. Validation of Primary Cell Cultures

For analysis of primary cells, low-passage cultured cells were centrifuged at 12,000 rpm (10 min); the supernatant was removed and replaced by 4% formaldehyde solution. After fixation (30 min), the formaldehyde was removed and cells were resuspended in 1% agarose in PBS followed by short cooling of the cell suspension at −20 °C. The solidified cell suspension was then placed in labelled cassettes, dehydrated, wax infiltrated, paraffin-embedded, and cut into 2–4 µm thick sections. Sections were mounted on TESPA-coated glass slides and were processed in routine staining and automated immunohistochemistry procedure using Dako Autostainer Link 48. (Agilent DAKO). For antibody details, see Table 2 (CNS tumors). OS morphology of all primary cell cultures was confirmed by morphological and immunocytochemical analysis performed by an experienced pathologist. The cells were examined for (1) cell morphology, signs of cell and nuclear atypia, presence of atypical mitoses, and osteoid formation (routine HE and Goldner stained sections); (2) lineage differentiation of the tumor cells (standardized immunocytochemical examination); and (3) exclusion of contaminating cell populations (standardized immunocytochemical examination), as described in detail in [32]. ES morphology was confirmed with the same procedure with additional staining with a CD99 antibody. 

### 4.6. Immune Cells Infiltration

Immune cell densities were evaluated in paraffin-embedded formalin fixed (FFPE) tumor tissue using immunohistochemistry and digital image analysis. After heat-induced antigen retrieval, FFPE tissue slides were stained by an automated immunostaining system (Thermo Fisher) using the antibodies in Table 3.

Stained slides were digitalized using a whole slide scanner (NanoZoomer, Hamamatsu Photonics, Hamamatsu, Japan) and analyzed by the open-source software QuPath [77]. At first, the tumor area was manually annotated followed by supervised machine-learning-based classification to separately annotate tumor tissue, non-tumor tissue (stroma and necrosis), and irrelevant structures (Appendix A). Cells were quantified by the cell detection tool followed by a threshold-based classification into positive and negative. Cell densities were assessed in technical duplicates (two tumor regions per sample) and expressed as number of positive cells per area.

### 4.7. Cell Proliferation

Cell growth was continuously monitored for at least 48 h using the xCELLigence RTCA MP instrument and the RTCA Data Analysis Software 1.0 (ACEA Biosciences, San Diego, CA, USA). Background impedance signal was measured with 50 μL cell culture medium per well. The final volume in a single well was adjusted to 100 μL cell culture medium by adding additional 50 μL medium containing 5000 tumor cells/well). After plating, impedance was routinely recorded in 15 min intervals. Then, 24 h after seeding, test compounds were added to the culture. All incubations were performed in 200 μL volume. For each compound and concentration, at least three replicates on an E-Plate 96 were performed, and each compound was analyzed at seven different concentrations (0.001, 0.01, 0.1, 1, 10, 100, and 300 μM); according concentration of compound free solvent (DMSO) were used as controls. After compound administration, impedance was measured every 15 min until the end of the experiment. The impedance signal was analyzed by normalizing data of each singe well to the last measurement after starting the treatment: CI (normalized) = CItime x/CInorm time (termed here as “normalized cell index”). This normalized cell index was used for graphical result representation and exported for further processing using Microsoft Excel and GraphPad Prism 7. Each Experiment was performed twice. 

### 4.8. Detection of the EWSR1-FLI1 Fusion

RNA isolation was conducted using the RNeasy Lipid Tissue Mini Kit (Qiagen, Hilden, Germany). RNA was converted to cDNA by using PrimeScript RT Reagent Kit with gDNA Eraser (Takara Bio Europe, Saint-Germain-en-Laye, France). Quality control was performed using a Bioanalyzer2100 (Agilent Technologies, Waldbronn, Germany). The EWSR1-FL1 fusion gene was amplified with primers 5′-TACAGCCAGCCTGTCCAG and 5′-GTGAGGATTGGTCGGTGTG, and the product was visualized on an agarose gel and analyzed by Sanger sequencing.

### 4.9. Patient History and Design of the Individual Treatment Protocol

A five-year-old girl with ES arising from the first left rib and spreading over the entire left hemithorax (see Appendix A) was treated according to the Ewing 2008 protocol. After five cycles of intensive induction chemotherapy including vincristine, ifosfamide, doxorubicin, and etoposide (VIDE), a significant tumor shrinkage was documented. After completion of the induction treatment (six cycles VIDE), the remaining tumor was resected incompletely due to diffuse infiltration of the left cervical neurovascular bundle. Only five weeks after the first surgery and two weeks after the first consolidation cycle with vincristine, actinomycin d, and cyclophosphamide (VAC), she presented with pain and paresis of the left arm. MRI revealed a local progressive tumor, invasively growing into the cervical spine. An immediate incomplete resection of the tumor (intraspinal part) was conducted, followed by two cycles of irinotecan and temozolomide and irradiation of the primary tumor site. Since primary refractory disease in ES is associated with extremely poor outcome, and no standardized curative salvage treatment is available, an individualized treatment concept with dinutuximab beta was evaluated for the patient [78]. Dinutuximab beta is licensed in Europe for first-line treatment in patients with high-risk neuroblastoma, a GD2 positive malignant pediatric cancer, with and without a combination of interleukin 2 (IL-2) and thus was used off label in this individualized treatment protocol. Without IL-2, dinutuximab beta is proven to be more tolerable regarding acute side effects like hypersensitivity reactions, fever, pain, and capillary leakage [79]. Thus, IL-2 was not included in the treatment protocol. The patient was considered to receive irinotecan (50 mg/m²/d intravenously infused over 30 to 90 min) and temozolomide (100 mg/m²/d orally; 1 h prior to irinotecan) on days one to five and dinutuximab beta (100 mg/m²/10 days; continuous intravenous infusion) on days two to eleven of a 21-day cycle in an outpatient setting. Supportive medication with gabapentin, morphine, paracetamol, and dimethindene was applied according to the product information of dinutuximab beta. The protocol was initially reviewed and approved by the Medical Service of the Health Funds (MDK). Repetitive evaluations were carried out every fourth cycle with regard to treatment efficacy and tolerability by the MDK. After being thoroughly informed about the goals of the individualized treatment, the off-label use of dinutuximab beta and the possible side effects, the custodians provided written informed consent. Formal approval of the local ethics committee for this study was not required, as this was a single case investigation.

### 4.10. Software

Graphical abstract was created using smart (http://smart.servier.com/).

### 4.11. Statistical analysis 

Difference in GD2 expression between tumor entities was calculated with one way ANOVA and Tukey’s multiple comparison test. GraphPad version 7 was used to perform the analysis.

## 5. Conclusions

GD2 is a potential therapeutic target in Ewing’s sarcoma and H3K27M-mutant DMG and can be addressed by monoclonal antibodies or by perturbation of the glycosphingolipids metabolism. Incorporating quantifiable methods of GD2 expression as a biomarker for enrolment on clinical trials of GD2-targeted agents may help to augment the likelihood of response to therapy. Further pre-clinical in vivo analyses are required to elucidate the mechanism of action of glucosylceramide synthase inhibitors on tumor growth and will facilitate the access of pediatric patients to innovative clinical studies

## Figures and Tables

**Figure 1 cancers-13-00520-f001:**
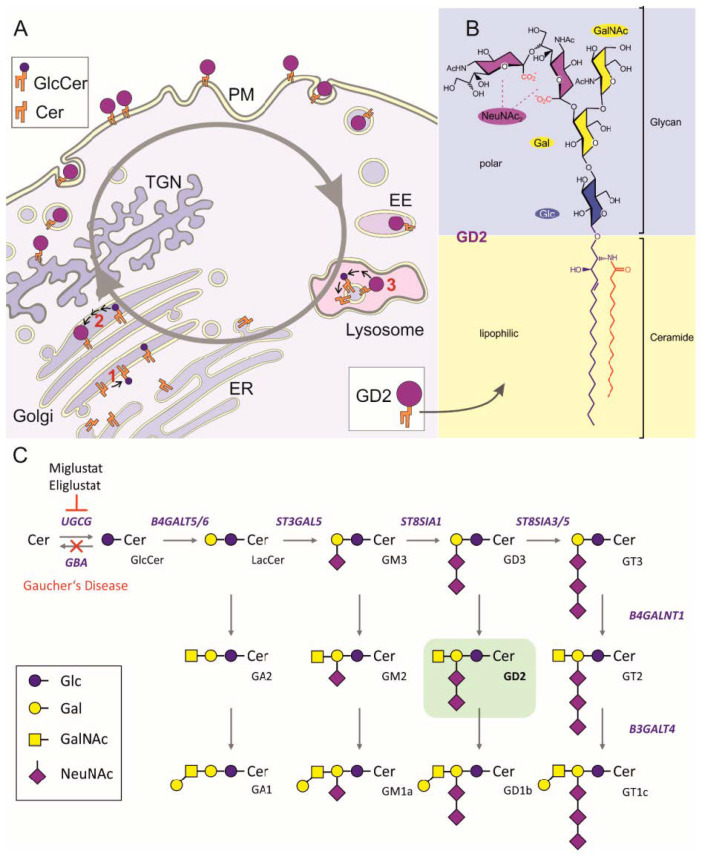
Cellular ganglioside metabolism and structure of GD2. (**A**). Cellular synthesis of the ganglioside GD2 starts with the synthesis of the lipid ceramide at the endoplasmic reticulum (ER). Transported to the Golgi apparatus (Golgi), it is converted by the glucosylceramide synthase (UGCG, 1) and subsequently luminal acting glycosyltransferases (2) into GD2 before it reaches the plasma membrane (PM) by vesicular transport from the trans Golgi network (TGN). Exposed to the outer leaflet of the PM, it may facilitate cell–cell contacts or modulate laterally the activity of plasma membrane proteins. Concentration of GD2 at the PM is further regulated by endocytic uptake of GD2, transport to early endosomes (EE), and stepwise lysosomal degradation into its sugar building blocks, free fatty acids, and sphingosine (3). The latter can be used again by the salvage pathway for production of ceramide at the ER. (**B**). Chemical structure of the ganglioside GD2 showing its lipophilic ceramide anchor, which is composed of a sphingoid base (blue) and a fatty acid (red), and the polar glycan head group. The latter is composed of a neutral core trisaccharide containing a terminal N-acetyl galactosamine (GalNAc) linked to a galactose (Gal), which is bound to the ceramide connected glucose (Glc). This trisaccharide is extended by two N-acetyl neuraminic acids (NeuNAc) attached to the galactosyl residue. (**C**). Combinatorial biosynthesis of gangliosides including GD2 (on green background) begins with the condensation of UDP-glucose and ceramide (Cer) to form glucosylceramide (GlcCer). This step is catalyzed by the glucosylceramide synthase (UGCG), which is inhibited by the small molecular compounds Eliglustat and Miglustat. By further glycosyl transferases (B4GALT5/6, ST3GAL5, ST8SIA1, and ST8SIA3/5) GlcCer is converted to lactosylceramide (LacCer) and the gangliosides GM3, GD3, and GT3, all of which are substrates for the GalNAc transferase B4GALNT1, leading to more complex structures GA2, GM2, GD2, and GT2. The latter may then be glycosylated by the galactosyl transferase B3GALT4 to produce GA1, GM1a, GD1b, and GT1c. GM1a, GD1b, and GT1c as well as GD1a (not included into scheme) are the main gangliosides of the brain. Lysosomal degradation of gangliosides is a stepwise process leading finally again to GlcCer, which is cleaved by lysosomal glucocerebrosidase (GBA). Deficiency of GBA causes the lysosmal storage disease Gaucher’s Disease. Substrate deprivation therapy with Eliglustat or Miglustat by inhibiting UGCG is one possibility to treat Gaucher’s Disease.

**Figure 2 cancers-13-00520-f002:**
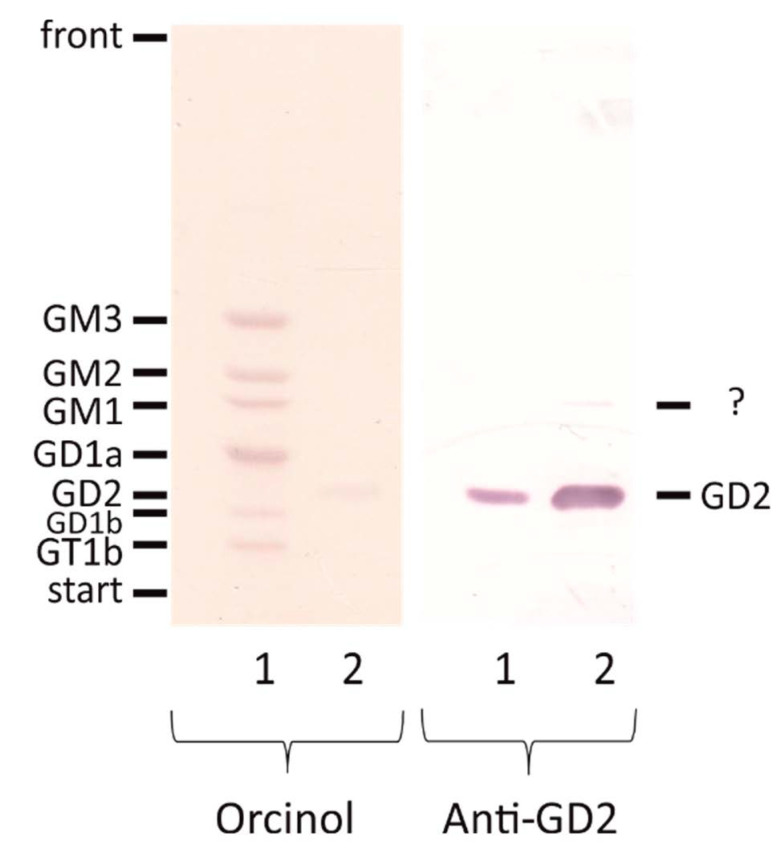
Validation of the anti-GD2 antibody used for flow cytometry. A ganglioside standard containing purified bovine brain cortex gangliosides and gangliosides GM3 and GM2 were loaded on lane 1 and standard ganglioside GD2 was loaded on lane 2. Gangliosides were separated by thin layer chromatography and subjected to immunostaining using the anti-GD2 antibody (right). Subsequently, staining was removed, and all gangliosides were stained with orcinol (left). The anti-GD2 antibody does not recognize any of the indicated gangliosides (GM3, GM2, GM1, GD1a, GD1b, GT1b) besides GD2. Although not visible by orcinol staining, GD2 is present in the ganglioside extract of bovine brain cortex. However, the anti-GD2 antibody recognizes an unknown compound, which is present in the GD2-standard (band marked with “?”).

**Figure 3 cancers-13-00520-f003:**
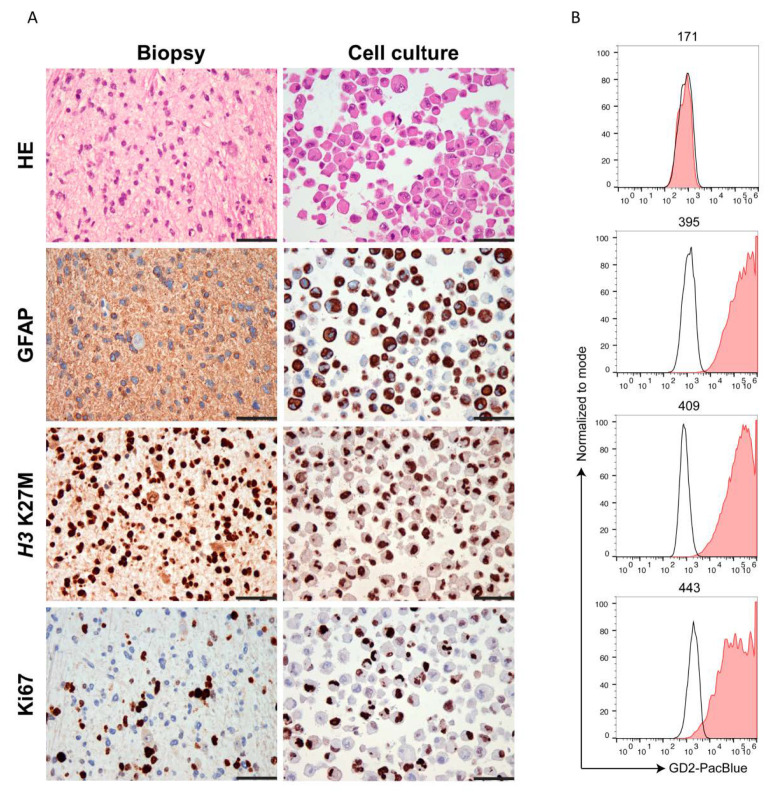
H3K27M-mutant DMG express high and homogeneous amount of GD2. (**A**). Validation of H3K27M-mutant DMG primary cells. Primary tumor cells of patient no. 395 were embedded in paraffin, stained with hematoxylin eosin (HE), and further analyzed by IHC with the indicated antibodies. The tumor used for the isolation of the primary cells was used as reference (biopsy, left panel). The H3K27M mutation was maintained in the cell culture. GFAP and Ki67 staining was similar between the biopsy and the primary cells. Scale bar: 50 µm. Similar results were obtained with sample 409. (**B**). Expression of GD2 in CNS tumors. Primary tumor cells were analyzed by flow cytometry with an anti-GD2 antibody. 171 cells derive from HGNET-BCOR, sample no. 395 and no. 409 from H3K27M-mutant DMG and no. 443 from DNET. Unstained samples were used as negative controls (black histogram).

**Figure 4 cancers-13-00520-f004:**
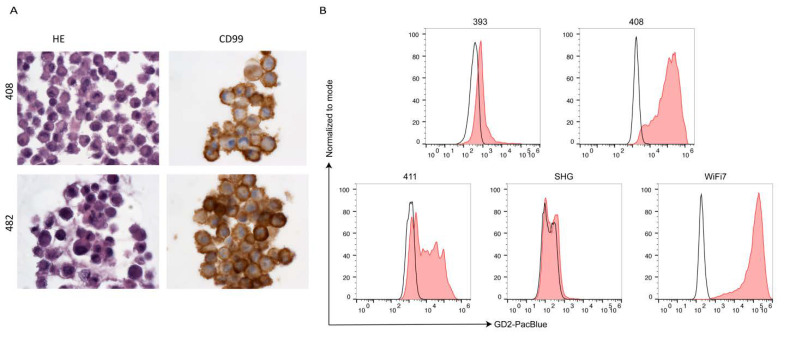
Some ES express high amounts of GD2. (**A**). Validation of ES primary cells. Primary tumor cells were embedded in paraffin and stained with hematoxylin eosin (HE) or analyzed by IHC with a CD99 antibody. Sample no. 408 was isolated from a metastasis in the spine and sample no. 482 from an intracranial metastasis of the same patient. Magnification 630X. (**B**). Expression of GD2 in ES. Primary ES tumor cells were analyzed by flow cytometry with an anti-GD2 antibody. Unstained samples were used as negative controls (black histogram).

**Figure 5 cancers-13-00520-f005:**
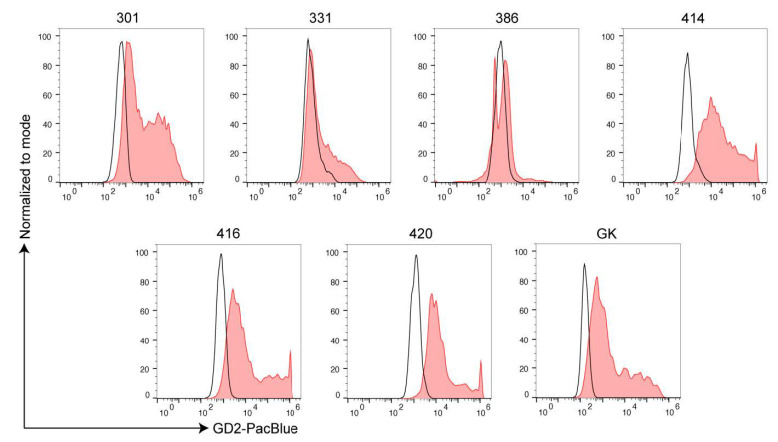
OS generally express low amount of GD2. Primary tumor cells of the indicated samples were analyzed by flow cytometry with an anti-GD2 antibody. Unstained samples were used as negative controls (black histogram).

**Figure 6 cancers-13-00520-f006:**
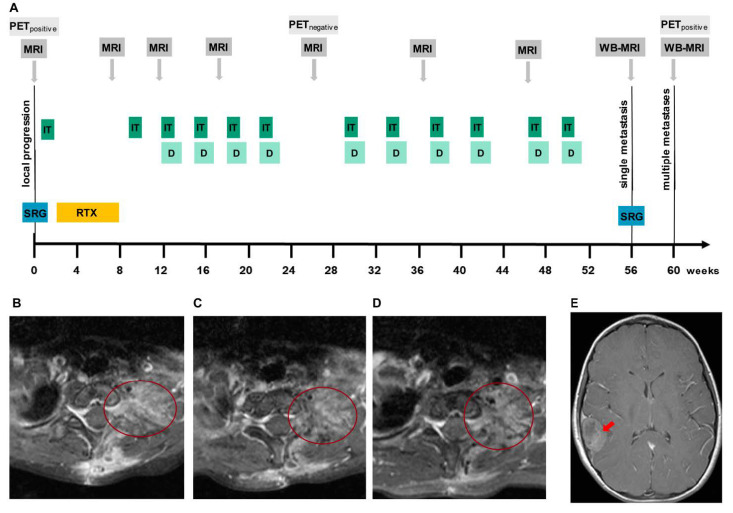
Clinical efficacy of dinutuximab in an ES patient. (**A**). Therapy protocol used for off-label application of dinutuximab beta. SRG: surgery; IT: irinotecan/temozolomide; RTX: irradiation of the primary tumor site. (**B**–**D**) Transversal T1-weighted fat-saturated turbo spin echo MRI sequences demonstrating post-therapeutic contrast enhancement at the upper left thoracic aperture lateral to the C7/Th1 neural foramen at week 7 (**B**), which was regressive at week 10 (**C**), with no signs of local progression at week 47 (**D**,**E**). Transversal T1-weighted contrast-enhanced spin echo MRI sequence showing a right temporal skull metastasis growing invasively into the brain at week 56 (red arrow).

**Figure 7 cancers-13-00520-f007:**
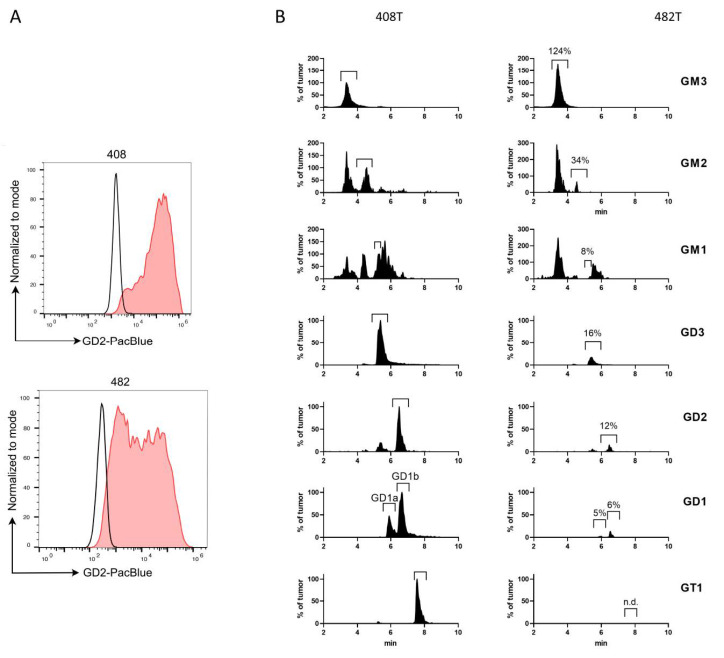
GD2 expression is reduced upon therapy with dinutuximab. (**A**). Expression of GD2 was analyzed by flow cytometry of primary tumor cells isolated before the dinutuximab beta therapy (no 408) or from the cranial metastasis developed under the dinutuximab beta therapy (no 482). (**B**) Liquid chromatography (HILIC)-coupled tandem mass spectrometry showing the composition of gangliosides in one sample of tumor 408T and 482T. 408T is the spinal metastasis that was used to isolate primary cells no 408. 482T is the intracranial metastasis used to isolate primary cells no 482 and developed under dinutuximab therapy (Figure 6E). Gangliosides were separated from neutral lipids prior to analysis. Tandem mass spectrometry detection was run in MRM mode. Retention times were compared to external standards (not shown). Total intensities were adjusted to corresponding total protein levels and normalized to the corresponding signals of the tumor. Percent (%) values given on the right side are corresponding to respective areas (and not peak heights).

**Figure 8 cancers-13-00520-f008:**
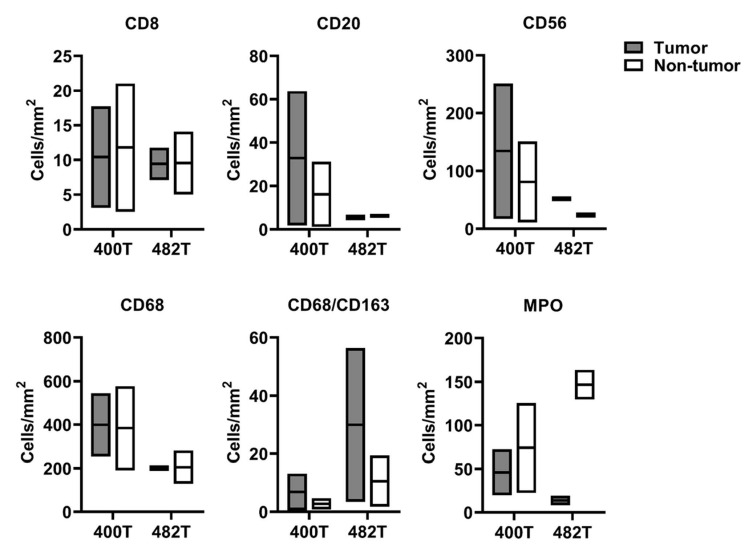
Infiltration of immune cells upon therapy with dinutuximab. Cell densities of cytotoxic T cells (CD8), B cells (CD20), NK cells (CD56), macrophages (CD68), CD163-positive (“M2”) macrophages, and neutrophils (MPO) were assessed by quantitative digital image analysis before (400T) and after (482T) dinutuximab therapy. Data is shown as min/max with mean for two tumor regions per time point. Representative stainings can be found in Appendix A.

**Figure 9 cancers-13-00520-f009:**
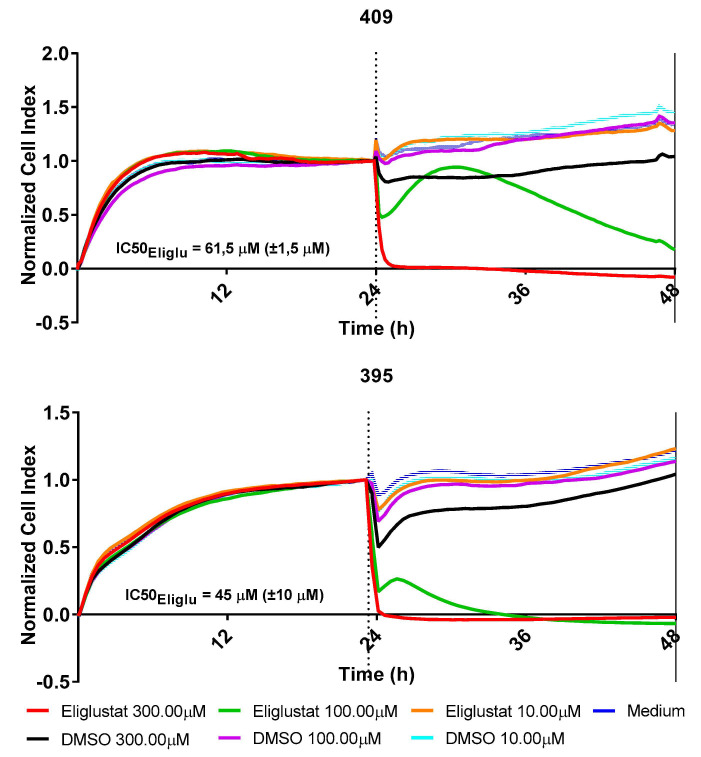
Eliglustat inhibits the proliferation of H3K27M-mutant DMG cells. Dose-response cell survival curves of H3K27M-mutant DMG primary tumor cells from two patients (no 395 and no 409) in response to eliglustat at the indicated concentrations. Due to clarity reasons, only the lowest non effective concentration (10 µM) is shown. DMSO was used as control; all experiments were performed two times and showed similar results. IC50_Eliglu_ indicates the mean half maximal inhibitory concentration of eliglustat and the range from two independent experiments for each sample.

**Table 1 cancers-13-00520-t001:** Primary tumor cells used in the study.

Sample	Gender	Age	Diagnosis	Primary Tumor or Metastasis	Localization	[%]
301	m	13	OS	Primary tumor	Distal femur	43.5
331	m	18	OS	Metastasis	Lung	10.9
386	f	16	OS	Metastasis	Lung	2.65
414	m	15	OS	Primary tumor	Proximal tibia	32.7
^1^416	m	15	OS	Primary tumor	Proximal tibia	24.9
^1^420	m	15	OS	Metastasis	Spine	17.4
GK	m	11	OS	Primary tumor	Proximal tibia	26.6
393	m	16	ES	Primary tumor	Clavicle	5.83
411	m	20	ES	Metastasis	Liver	57.5
^2^408	f	5	ES	Metastasis	Spine	91
^2^482	f	6	ES	Metastasis	Intracranial	59.3
WiFi7	m	28	ES	Primary tumor	Femur	98.4
SHG	m	50	ES	Metastasis	pleural effusion	0.9
443	m	9	DNET	Primary tumor	CNS	96.4
171	m	7	HGNET-BCOR	Metastasis	Skull	0
395	m	16	H3K27M-mutant DMG	Primary tumor	CNS	98.1
409	m	7	H3K27M-mutant DMG	Primary tumor	CNS	99

^1^ same patient: 416 is at first diagnosis, no. 420 is a metastasis one months after the first diagnosis. ^2^ same patient, before (408) and after (482) dinutuximab therapy. % indicates the percent of the cell population expressing GD2 and was used for thescoring of GD2 expression: ≤5% negative, 5–30% low, 30–70% intermediate, 70–100% high.

**Table 2 cancers-13-00520-t002:** Antibodies used in IHC analysis of CNS tumors.

Antibody	Dilution	Host	Clone	Provider	No.	Incubation Time (min)
anti-GFAP	RTU	rabbit	Pk	Dako	IR524	20
anti-Ki-67	RTU	mouse	MIB-1	Dako	IR626	20
anti-Synaptophysin	RTU	mouse	DAK-Synap	Dako	IR660	20
anti-Neurofilament	RTU	mouse	2F-11	Dako	IR607	20
anti-H3-K27-M	1/500	rabbit	Pk	Millipore	ABE419	20

RTU = ready to use.

**Table 3 cancers-13-00520-t003:** Antibodies used for the analysis of infiltrating immune cells.

Antibody	Dilution	Host	Clone	Provider	No.	Incubation Time (min)
CD8	RTU	mouse	C8/144B	DAKO	IR623	20
CD20	RTU	mouse	L26	DAKO	IR604	20
MPO	RTU	rabbit	polyclonal	DAKO	IR511	20
CD68	RTU	mouse	PG-M1	DAKO	IR613	20
CD163	1/200	mouse	10D6	Leica	CD163-L-CE	20
CD56	RTU	mouse	123C3	DAKO	IR628	20

## Data Availability

The data presented in this study are available on request from the corresponding author.

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
