# Peer review of "Exploiting Gangliosides for the Therapy of Ewing’s Sarcoma and H3K27M-Mutant Diffuse Midline Glioma"

_cancers, 2021, doi:10.3390/cancers13030520_

Round 1
Reviewer 1 Report
Study of the effectiveness of anti-GD2 antibodies to treat GD2-expressing tumors other than neuroblastoma is not a new concept but is one that needs more interrogation. There are however a number of points the authors need to address.
- Conclusions given by authors are a bit grandiose considering that only a single patient was treated.
- Introduction, don’t really mention work done on melanoma and osteosarcomas with anti-GD2 antibodies in addition to NBs (eg. Navid, F., Santana, V. M., & Barfield, R. C. (2010). Anti-GD2 antibody therapy for GD2-expressing tumors. Current cancer drug targets, 10(2), 200–209. https://doi.org/10.2174/156800910791054167). Not a new or novel concept.
- Supplemental figure 1 can be deleted as it doesn’t vary much from Figure 2; can just say results were similar to those obtained for H3K27M-mutant DMG 409.
- 7, line 1: indicate the ES patient 408 is 6 yrs but in table 1 is listed at 4 yrs.
- In Figures S6 and 7 need to include number, as listed in table 1, of tumor studied.
- When using eliglustat did cell growth resume upon removal of the lesser concentrations? What effect did it have on cell ganglioside expression? Should also note that its approval for treatment of Gaucher’s disease is to limit production of glucosylceramide that cells cannot degrade, but could potentially use as a precursor for synthesis of gangliosides needed for neural cell function. Consideration should be given to what might happen if it is used as a cancer treatment and ganglioside concentrations are significantly altered.
- Figure 8 shows results for just 3 different concentrations of drug while the methods section indicates that 7 different ones were used. Calculation of the IC50s might have been more accurate if a few of the concentrations tested were between 10 and 100µM.
- Indicate the connection between GD2 levels and the H3F3A K27M and the HIST1H3 K27M mutations.
- Some editing is needed: eg: line 143, should delete therefore; line 149, commercial should read commercially; line 206, provide best possibly should read provide the best possible; line 240, scull should read skull; line 255, has should read had; and line 370, what is meant by no less meaningful treatment? Line 546, acetonitril should read acetonitrile; line 563, cutted should read cut.
Author Response
Study of the effectiveness of anti-GD2 antibodies to treat GD2-expressing tumors other than neuroblastoma is not a new concept but is one that needs more interrogation. There are however a number of points the authors need to address.
- Conclusions given by authors are a bit grandiose considering that only a single patient was treated.
We thank the Reviewer for this advice. We changed the conclusion to:
“GD2 is a potential therapeutic target in Ewing’s sarcoma and H3K27M-mutant DMG and can be addressed by monoclonal antibodies or by perturbation of the glycosphingolipids metabolism. Incorporating quantifiable methods of GD2 expression as a biomarker for enrolment on clinical trials of GD2-targeted agents may help to augment the likelihood of response to therapy. Further pre-clinical in vivo analyses are required to elucidate the mechanism of action of glucosylceramide synthase inhibitors on tumor growth and will facilitate the access of pediatric patients to innovative clinical studies”
- Introduction, don’t really mention work done on melanoma and osteosarcomas with anti-GD2 antibodies in addition to NBs (eg. Navid, F., Santana, V. M., & Barfield, R. C. (2010). Anti-GD2 antibody therapy for GD2-expressing tumors. Current cancer drug targets, 10(2), 200–209. https://doi.org/10.2174/156800910791054167). Not a new or novel concept.
We addressed the reviewer’s opinion by adding the following to the Introduction (line 109 ff):
“In OS, GD2 has been suggested to plays a role in chemotherapy resistance and tumor progression [15]. Phase I Clinical trials conducted with melanoma and osteosarcoma patients showed response to anti-GD2 antibody treatment but only in a fraction of the treated patients [16] [17]. This indicates the significance of a proper stratification to identify eligible patients for the anti-GD2 treatment.“ Moreover we included the Ref indicated by the reviewer (Ref 3)
Supplemental figure 1 can be deleted as it doesn’t vary much from Figure 2; can just say results were similar to those obtained for H3K27M-mutant DMG 409.
We have eliminated Supplemental Figure 1 according to the reviewer’s comments
- 7, line 1: indicate the ES patient 408 is 6 yrs but in table 1 is listed at 4 yrs.
Patient age at time of diagnosis was 4 yrs. At time of progression (resection of the spinal metastasis, cells named as 408) and beginning of dinutuximab treatment the patient was 5 yrs., not 4 yrs. as mentioned in table 1. The cranial metastasis was detected at the age of 6 yrs. To avoid misunderstanding we changed the age in text, and table 1 to the correct age (5 yrs.) at moment of resection and treatment initiation and to 6 yrs (Table 1) concerning the appearance of the metastasis. We apologize for this mistake.
- In Figures S6 and 7 need to include number, as listed in table 1, of tumor studied.
We have added the number
- When using eliglustat did cell growth resume upon removal of the lesser concentrations? What effect did it have on cell ganglioside expression? Should also note that its approval for treatment of Gaucher’s disease is to limit production of glucosylceramide that cells cannot degrade, but could potentially use as a precursor for synthesis of gangliosides needed for neural cell function. Consideration should be given to what might happen if it is used as a cancer treatment and ganglioside concentrations are significantly altered.
We did not analyse the cell growth under this condition and ganglioside expression was not analyzed so far. Recently, eliglustat was linked to reductions in glycosphingolipids in RM-9 tumor bearing mice. We have added this comment in the discussion at line 565
To our knowledge eliglustat and miglustat have not been used for cancer treatment. We cannot anticipate what side effects will result upon treatment without underlying enzymatic dysfunction. This must be addressed in early phase clinical trials for oncological patients without Gaucher’s disease or Niemann-Pick disease. However, in a mouse model without enzymatic dysfunction it did not induce relevant neurotoxicity. We discussed this point starting from line 581:
„ In the context of lipid storage disorder diseases the treatment with glucosylceramide synthase inhibitors is generally well tolerated with major toxicities related to the gastrointestinal tract that can be managed with a particular diet. Whether the treatment has side effects in patients without underlying enzymatic dysfunction needs to be clarified in further clinical trials. This could be important because gangliosides are needed for neural cell function. However, neurotoxicity under treatment with eliglustat was not described in C57BL/6N mice without enzymatic dysfunction [71]”.
- Figure 8 shows results for just 3 different concentrations of drug while the methods section indicates that 7 different ones were used. Calculation of the IC50s might have been more accurate if a few of the concentrations tested were between 10 and 100µM.
We appreciate the reviewers notice about this inaccuracy. We performed the Experiment with 7 different concentrations as described in material and methods (1 nM, 10 nM, 100 nM, 1 µM, 10 µM, 100 µM and 300 µM). Concentrations between 1nM and 10 mM did not show cytotoxic effects on the cells. For clarification we add at line 386: “, while cell growth at lower concentration (≤ 10 mM) was still observed”. Due to clarity reasons we show only the lowest non effective concentration. We addressed this by changing the figure legend. As suggested by the reviewer, additional concentrations between 10 and 100 µM would be necessary to calculate the IC50 more accurately, therefore we add IC50 of “about” at line 385.
- Indicate the connection between GD2 levels and the H3F3A K27M and the HIST1H3 K27M mutations.
We have added at line 442: “GD2 overexpression in K27M mutants is related to transcriptional perturbation resulting from the histone mutation which induces upregulation of ganglioside synthesis enzymes [14]”
- Some editing is needed: eg: line 143, should delete therefore; line 149, commercial should read commercially; line 206, provide best possibly should read provide the best possible; line 240, scull should read skull; line 255, has should read had; and line 370, what is meant by no less meaningful treatment? Line 546, acetonitril should read acetonitrile; line 563, cutted should read cut.
All editing has been done according to the suggestion
Reviewer 2 Report
A central point of this paper, that H3K27M-mutant DMG might benefit from treatment with inhibitors of glucosylceramide synthase as currently used for treating children with Niemann Pick or Gaucher, is of major importance. The authors idea of using eliglustat in cancers where GD2 plays a pathophysiologic role has potential to be transformative to clinical practice. This is an excellent paper of great worth.
Language use is generally good. Many minor points listed below do require attention though. Two major points absolutely require addressing: 1] Given the reported patient’s dire situation and the authors’ in vitro findings, why was an FDA and EMA approved drug, eliglustat, not given ? This must be clearly addressed. 2] The entire Discussion section must be re-written. This should not require more than 8 hours. Division into appropriately titled subsections and breaking up run-on paragraphs are necessary.
Line 64, would the authors think it useful to readers to provide a figure of the basic chemical structure of these elements of GD2 ? Providing this early in the Introduction will help make GD2 less abstract to clinician/researchers who have only vague memories of what ceramide or “sialic acids” are. It is only on page 11 that the structure of GD2 is schematically given.
Line 84, it is unclear to me how an antibody, or anything else, found in CSF is going to tumor tissue within brain parenchyma.
Line 87, “Beyond antibody-based approaches…” might be better as new paragraph.
Line 94, “ might benefit from treatment with inhibitors of glucosylceramide synthase …” might be better as “ might benefit from treatment with small molecule inhibitors of glucosylceramide synthase…” ?
Regarding CD99, I didn’t understand the significance of tissue staining for this. Maybe the authors should add a short paragraph on implications of staining or non-stainning for CD99.
Given the incidence of glioblastoma (GB), would the authors like to give some IHC high mag. images of this ? Although the authors state that U-87 had low GD2 expression [line 152], GD2 IHC images of human GB biopsy would help us evaluate further exploration of potential for eliglustat in GB. Please see references 1 to 9 below for others who are exploring this, and refer to their work if the authors feel these are relevant.
Line 184, “DMG and some ES highly express GD2 and are candidates for therapies addressing GD2.” is an assumption without backing. “Highly expressing” refers to a moment in time. That can change over time. Also a pathology mediating molecule need not be expressed in large amounts to be a worthwhile target in treatment. EGFR is in NSCLC but that need not generalize to other markers in other cancers.
Line 195, I would like to see those results re. INFORM platform.
Line 212, we would benefit from listing of specific O2 sat nadir %.
Line 214, “Pain was adequately controlled…” with ? Given the nature of this case, listing ancillary meds like this is important.
Line 253, “The EWSR1-FL1 fusion” would benefit from an explanation. I don’t know the significance of this. If the authors do, they should tell us.
Line 271, “ By contrast, the overall neutrophil density was higher, but the number of tumor-associated neutrophils was lower after dinutuximab therapy. “ is unclear. What are the authors saying here ? Also these leukocyte changes must be discussed in Discussion section. What significance do the authors attach to these changes ? The potentially tumor-enhancing role of neutrophils in various cancers is well-known, as are measures to decrease these contributions. See references 10-15 below as examples.
Figure 8 must be re-written, probably separated into two figures [a fig. 8 and fig.9]. The legend is grossly inadequate. Many of the abbreviations in the figure are not listed in legend. Also the thousand fold difference between IC50 of eliglustat at glucosylceramide synthase [nM] and IC50 cell index [microM] must be mentioned and discussed.
Typos and wording change suggestions:
on line 103, “seeparated”
on line 106, “a antibody-positive band”
on line 131 “to the patients” [ should read: to the patients’ ]
on line 135 - “to a lesser extend”
on line 138 “H3K27M-mutant DMG samples had a very expression of GD2 138 with all cells being strongly positive…”
Wrong preposition, line 184 “treatment of the Ewing 2008 treatment protocol”
Line 195, better to start new paragraph.
Line 199, better to start new paragraph.
Line 202 should read “were considered to…”
Line 293 “ synthesis by eliglustat, a…” should read “synthesis with eliglustat, a...” or “ synthesis by using eliglustat, a...” Also “in vitro” must be mentioned at onset of this subject, line 289. At first mention of eliglustat the trade name should be given in parentheses (Cerdelga®).
The MOA of eliglustat in treating Gaucher must be given with references. is a specific inhibitor of glucosylceramide synthase (IC50=10 ng/mL) and acts as a substrate reduction therapy for GD1 by reducing the production of GL-1.
Neutrophils as MDSC [myeloid derived suppressor cells] must be discussed in view of the authors’ findings.
______________________________________________
1: Marx S, Wilken F, Wagner I, Marx M, Troschke-Meurer S, Zumpe M, Bien-Moeller S, Weidemeier M, Baldauf J, Fleck SK, Rauch BH, Schroeder HWS, Lode H, Siebert N. GD2 targeting by dinutuximab beta is a promising immunotherapeutic approach against malignant glioma. J Neurooncol. 2020;147(3):577-585. doi:10.1007/s11060-020-03470-3.
2: Murty S, Haile ST, Beinat C, Aalipour A, Alam IS, Murty T, Shaffer TM, Patel CB, Graves EE, Mackall CL, Gambhir SS. Intravital imaging reveals synergistic effect of CAR T-cells and radiation therapy in a preclinical immunocompetent glioblastoma model. Oncoimmunology. 2020;9(1):1757360. doi:10.1080/2162402X.2020.1757360.
3: Fleurence J, Bahri M, Fougeray S, Faraj S, Vermeulen S, Pinault E, Geraldo F, Oliver L, Véziers J, Marquet P, Rabé M, Gratas C, Vallette F, Pecqueur C, Paris F, Birklé S. Impairing temozolomide resistance driven by glioma stem-like cells with adjuvant immunotherapy targeting O-acetyl GD2 ganglioside. Int J Cancer. 2020;146(2):424-438. doi:10.1002/ijc.32533.
4: Golinelli G, Grisendi G, Prapa M, Bestagno M, Spano C, Rossignoli F, Bambi F, Sardi I, Cellini M, Horwitz EM, Feletti A, Pavesi G, Dominici M. Targeting GD2-positive glioblastoma by chimeric antigen receptor empowered mesenchymal progenitors. Cancer Gene Ther. 2020;27(7-8):558-570. doi:10.1038/s41417-018-0062-x.
5: Fleurence J, Cochonneau D, Fougeray S, Oliver L, Geraldo F, Terme M, Dorvillius M, Loussouarn D, Vallette F, Paris F, Birklé S. Targeting and killing glioblastoma with monoclonal antibody to O-acetyl GD2 ganglioside. Oncotarget. 2016;7(27):41172-41185. doi:10.18632/oncotarget.9226.
6: Woo SR, Oh YT, An JY, Kang BG, Nam DH, Joo KM. Glioblastoma specific antigens, GD2 and CD90, are not involved in cancer stemness. Anat Cell Biol. 2015;48(1):44-53. doi:10.5115/acb.2015.48.1.44.
7: Mennel HD, Lell B. Ganglioside (GD2) expression and intermediary filaments in astrocytic tumors. Clin Neuropathol. 2005;24(1):13-8.
8: Arbit E, Cheung NK, Yeh SD, Daghighian F, Zhang JJ, Cordon-Cardo C, Pentlow K, Canete A, Finn R, Larson SM. Quantitative studies of monoclonal antibody targeting to disialoganglioside GD2 in human brain tumors. Eur J Nucl Med. 1995;22(5):419-26. doi:10.1007/BF00839056.
9: Longee DC, Wikstrand CJ, Månsson JE, He X, Fuller GN, Bigner SH, Fredman P, Svennerholm L, Bigner DD. Disialoganglioside GD2 in human neuroectodermal tumor cell lines and gliomas. Acta Neuropathol. 1991;82(1):45-54. doi:10.1007/BF00310922.
10: Kast RE. Paths for Improving Bevacizumab Available in 2018: The ADZT Regimen for Better Glioblastoma Treatment. Med Sci (Basel). 2018;6(4):84. doi:10.3390/medsci6040084.
11: Wu L, Zhang XH. Tumor-Associated Neutrophils and Macrophages-Heterogenous but Not Chaotic. Front Immunol. 2020;11:553967. doi:10.3389/fimmu.2020.553967.
12: Theron AJ, Steel HC, Rapoport BL, Anderson R. Contrasting Immunopathogenic and Therapeutic Roles of Granulocyte Colony-Stimulating Factor in Cancer. Pharmaceuticals (Basel). 2020;13(11):406. doi:10.3390/ph13110406.
13: Hajizadeh F, Aghebati Maleki L, Alexander M, Mikhailova MV, Masjedi A, Ahmadpour M, Hashemi V, Jadidi-Niaragh F. Tumor-associated neutrophils as new players in immunosuppressive process of the tumor microenvironment in breast cancer. Life Sci. 2021;264:118699. doi:10.1016/j.lfs.2020.118699.
14: Wu M, Ma M, Tan Z, Zheng H, Liu X. Neutrophil: A New Player in Metastatic Cancers. Front Immunol. 2020;11:565165. doi:10.3389/fimmu.2020.565165.
Author Response
A central point of this paper, that H3K27M-mutant DMG might benefit from treatment with inhibitors of glucosylceramide synthase as currently used for treating children with Niemann Pick or Gaucher, is of major importance. The authors idea of using eliglustat in cancers where GD2 plays a pathophysiologic role has potential to be transformative to clinical practice. This is an excellent paper of great worth.
We thank the Reviewer for this appreciation of our work.
Language use is generally good. Many minor points listed below do require attention though. Two major points absolutely require addressing:
1] Given the reported patient’s dire situation and the authors’ in vitro findings, why was an FDA and EMA approved drug, eliglustat, not given? This must be clearly addressed.
While generating the presented data the patient was under the standard of care second line treatment with radiotherapy and oral chemotherapy. As progression of disease is anticipated, we applied for a review of an individualized salvage treatment with Miglustat by the Medical Service of the Health Funds (MDK) and the ethics committee. We hope to be able to offer the patient a therapy with Miglustat in near future.
2] The entire Discussion section must be re-written. This should not require more than 8 hours. Division into appropriately titled subsections and breaking up run-on paragraphs are necessary.
We have divided the discussion in subsections correlated to the results part. We have commented the concentration of eliglustat. We have commented the immune cell infiltration (see below for details).
Line 64, would the authors think it useful to readers to provide a figure of the basic chemical structure of these elements of GD2 ? Providing this early in the Introduction will help make GD2 less abstract to clinician/researchers who have only vague memories of what ceramide or “sialic acids” are. It is only on page 11 that the structure of GD2 is schematically given.
Thank you for this suggestion! We have introduced Figure 1, which shows the cellular ganglioside metabolism and structure of GD2.
Line 84, it is unclear to me how an antibody, or anything else, found in CSF is going to tumor tissue within brain parenchyma.
Thank you for this comment. Since concentrations of most drugs rapidly decrease to insignificant levels at few millimeters from ependymal surface, intrathecally applied substances are considered to have limited effect on parenchymal masses. We refined the sentence as follows:”… To overcome this limitation application of antibodies either into the cerebrospinal compartment in case of leptomeningeal disease or directly through the interstitial spaces of the CNS into parenchymal masses are reasonable approaches, having been subject of early clinical trials…” (line 121 ff.)
Line 87, “Beyond antibody-based approaches…” might be better as new paragraph.
Done
Line 94, “ might benefit from treatment with inhibitors of glucosylceramide synthase …” might be better as “ might benefit from treatment with small molecule inhibitors of glucosylceramide synthase…” ?
We have changed this according to the reviewer comment.
Regarding CD99, I didn’t understand the significance of tissue staining for this. Maybe the authors should add a short paragraph on implications of staining or non-stainning for CD99.
We have added that CD99 „ is used in routine diagnostic to distinguish the Ewing family of tumors from other tumors of similar histological appearance” (line 224).
Given the incidence of glioblastoma (GB), would the authors like to give some IHC high mag. images of this? Although the authors state that U-87 had low GD2 expression [line 152], GD2 IHC images of human GB biopsy would help us evaluate further exploration of potential for eliglustat in GB. Please see references 1 to 9 below for others who are exploring this, and refer to their work if the authors feel these are relevant.
We thank the reviewer for this excellent idea. Glioblastoma is not a common tumor entity in pediatric oncology. Therefore, the access to adequate tumor samples is limited for us and no further analysis could be performed on GB biopsies. In addition, IHC GD2 staining is up to our knowledge not sufficiently established. We are currently working on this with our collaborating pathologists and will than perform expression screenings with IHC for other tumors (e.g. GB from adult patients).
Line 184, “DMG and some ES highly express GD2 and are candidates for therapies addressing GD2.” is an assumption without backing. “Highly expressing” refers to a moment in time. That can change over time. Also a pathology mediating molecule need not be expressed in large amounts to be a worthwhile target in treatment. EGFR is in NSCLC but that need not generalize to other markers in other cancers.
As pointed out by the reviewer, we generally only have a snapshot of a tumor, which makes it difficult to generalize whether a target is a good target. This problem can be partially overcome by using liquid biopsy, but liquid biopsy cannot describe the heterogeneity of a tumor. In real life, we treat patients based on the assumption that a target remains constant in time unless otherwise shown. Since heterogeneity is one of the factors that clearly plays a role in resistance to therapy, we changed the sentence to: “DMG and some ES show uniform and high GD2 expression”
Line 195, I would like to see those results re. INFORM platform.
The patient’s tumor tissue as well as peripheral blood were analyzed in the framework of the INFORM registry. Whole genome sequencing (Agilent SureSelect v5 and low coverage, copy number analysis) of tumor tissue and peripheral blood was performed as well as DNA methylation analysis (Illumina Array) of tumor tissue. Transcriptome analysis could not be carried out due to technical issues. No actionable target was elucidated. Further data were not provided by the INFORM registry.
Line 212, we would benefit from listing of specific O2 sat nadir %.
We include the lowest level of oxygen saturation in the sentence as follows: “…Despite routine supportive medication, the patient presented with fever and mild oxygen desaturation with the lowest blood oxygen level of 89% requiring oxygen therapy for few days…”
Line 214, “Pain was adequately controlled…” with ? Given the nature of this case, listing ancillary meds like this is important.
We changed this to “As supportive standard of care, pain was adequately controlled with gabapentin, acetaminophen and morphine. In the later cycles morphine was no longer necessary.”
Line 253, “The EWSR1-FL1 fusion” would benefit from an explanation. I don’t know the significance of this. If the authors do, they should tell us.
EWSR1-FL1 should be EWSR1-FLI1. We apology for this error. Ewing’s sarcoma is genetically defined by a translocation that involves the EWSR1 gene and in most case the FLI1 gene and the presence of the fusion confirms a diagnosis of ES. We have added this information at line 339.
Line 271, “ By contrast, the overall neutrophil density was higher, but the number of tumor-associated neutrophils was lower after dinutuximab therapy. “ is unclear. What are the authors saying here ? Also these leukocyte changes must be discussed in Discussion section. What significance do the authors attach to these changes ? The potentially tumor-enhancing role of neutrophils in various cancers is well-known, as are measures to decrease these contributions. See references 10-15 below as examples.
The overall neutrophil density (cells per square millimeter) was higher in the post-treatment tumor specimen, however, we analyzed tumor and non-tumor (i.e. stroma and necrosis) cell densities separately and found an increased PMN density in non-tumor tissue but a decreased density in tumor tissue. We have explained better the definition of tumor and non-tumor in the results (line 355 ff.).
We have discussed the leukocytes changes at line 521. “The reallocation of neutrophils in post-treatment specimen with increased numbers in stromal and necrotic areas might also suggest reduced ADCC activity, since the destruction of opsonized cells requires a direct contact between tumor cells and neutrophils [61]. On the other hand, the immunohistochemical markers used in this study cannot differentiate between anti- and pro-tumorigenic myeloid cells, including myeloid-derived suppressor cells (MDSC) and certain TAN and TAM subsets that promote tumor growth and metastasis amongst others by suppression of adaptive immune functions and extracellular matrix remodeling [62]. We found only low densities of M2-like macrophages, but their slight increase following antibody treatment might be an indication for effective ADCP since ongoing phagocytosis alters macrophage function and phenotype into this direction [63].”
Figure 8 must be re-written, probably separated into two figures [a fig. 8 and fig.9]. The legend is grossly inadequate. Many of the abbreviations in the figure are not listed in legend.
The figure was separated, and figure legends were edited for further clarification. Figure 8A is now part of Figure 1.
Also the thousand fold difference between IC50 of eliglustat at glucosylceramide synthase [nM] and IC50 cell index [microM] must be mentioned and discussed.
Thank you for this legitimate remark. Jody Vykoukal et al. (Reference nr 71 in the publication) studied the effect of eliglustat in a prostatic cancer cell line (RM-9) and in a corresponding xenograft in vivo mouse model. They showed that eliglustat at 128 mM has a cytotoxic effect on RM-9 cells. To verify if this approximately thousand-fold higher IC50 value (IC50 of eliglustat at glucosylceramide synthase is 24 nM) can induce a clinically significant therapeutic effect on in vivo tumor growth, they treated xenograft RM-9 C57BL/6N mice with Eliglustat. Remarkably, eliglustat suppressed tumor growth in a therapeutic dosage of 20mg/kg. Our data are in full support of this, since we could observe an IC50 of 45-61.5 mM.
We introduced this thought in the discussion: „ In line with our data (IC50 45-61.5 mM), the cytotoxic concentration of eliglustat required to inhibit the prostatic cancer cell line RM-9 in vitro was 128 mM. Despite the high amount of eliglustat needed to induce this in vitro cytotoxicity, RM-9 tumor growth in vivo was still suppressed by using eliglustat at a therapeutic dosage and tumor tissues showed reductions in glycosphingolipids” (from line 565) .
Typos and wording change suggestions:
- on line 103, “seeparated”
- on line 106, “a antibody-positive band”
- on line 131 “to the patients” [ should read: to the patients’ ]
- on line 135 - “to a lesser extend”
- on line 138 “H3K27M-mutant DMG samples had a very expression of GD2 138 with all cells being strongly positive…”
- Wrong preposition, line 184 “treatment of the Ewing 2008 treatment protocol”
- Line 195, better to start new paragraph.
- Line 199, better to start new paragraph.
- Line 202 should read “were considered to…”
- Line 293 “ synthesis by eliglustat, a…” should read “synthesis with eliglustat, a...” or “ synthesis by using eliglustat, a...” Also “in vitro” must be mentioned at onset of this subject, line 289. At first mention of eliglustat the trade name should be given in parentheses (Cerdelga®).
All Typos and wording change suggestions have been accepted
The MOA of eliglustat in treating Gaucher must be given with references. is a specific inhibitor of glucosylceramide synthase (IC50=10 ng/mL) and acts as a substrate reduction therapy for GD1 by reducing the production of GL-1.
We have added this information in the results at line 383
Neutrophils as MDSC [myeloid derived suppressor cells] must be discussed in view of the authors’ findings.
We have discussed the MDSC from line 524 “On the other hand, the immunohistochemical markers used in this study cannot differentiate between anti- and pro-tumorigenic myeloid cells, including myeloid-derived suppressor cells (MDSC) and certain TAN and TAM subsets that promote tumor growth and metastasis amongst others by suppression of adaptive immune functions and extracellular matrix remodeling [54].”
Reviewer 3 Report
Brief Summary: The aim of the study by Wingerter and El Malki et al., was to assess the possibility of targeting the ganglioside GD2 lipid for the treatment of pediatric tumor entities, including Ewing’s sarcoma and H3K27M-mutant diffuse midline glioma (DMG). After validating the specificity of an ant-GD2 antibody for the use in flow cytometry, the authors assessed its expression in tumor cells derived from osteosarcoma, high-grade neuroepithelial tumor with BCOR alteration, Ewing’s sarcoma, dysembryoplastic neuroepithelial tumor and H3K27M-mutant DMG primary or metastatic tumors. They show that osteosarcoma cells were weakly positive for GD2, high-grade neuroepithelial tumor with BCOR alteration cells were GD2 negative and the dysembryoplastic neuroepithelial tumor cells strongly positive for GD2. Ewing’s sarcoma and H3K27M-mutant DMG cells were also strongly positive for GD2 expression. They also validated the origin of the H2K27M-mutant DMG and Ewing’s sarcoma cells by immunohistological analyses comparing the cultured cells with tumor biopsies. A case study of a Ewing’s sarcoma patient treated with a therapeutic anti-GD2 antibody (dinutuximab beta), showed a that the addition of anti-GD2 antibody to irinotecan/temozolomide therapy led to disease control for up to 12 months. Cells isolated before and after dinutuximab beta treatment revealed reduction of GD2, increase of GM3 glycolipid expression and a reduction immune cell infiltration in the cells from the post-dinutuximab beta treated tumor. Finally, the authors show that inhibition of sphingolipid synthesis by the agent eliglustat, decreases the in vitro proliferation of two H2K27M-mutant DMG-derived tumor cell lines. This study presents interesting data on potential approach to treat some pediatric tumors with high GD2 expression but could benefit from some additional analyses/clarifications and would definitely require incorporation of statistical analyses to support the author’s claims.
Strengths of the study:
- Identification of a potentially new biomarker, GD2, for aggressive pediatric tumors.
- Description of approaches to treat aggressive pediatric tumors via GD2 targeting.
- Inclusion of human case study where an anti-GD2 therapeutic approach was used showcases feasible clinical translation for Ewing’s sarcoma treatment.
Weaknesses of the study:
- Study is mainly descriptive, not providing any mechanistical insight on the action and importance of GD2 in H2K27M-mutant DMG or Ewing’s sarcoma.
- Lack of a more in-depth analyses of presented data hinder the significance of the study and results.
- Lack of statistical analyses throughout the manuscript.
Specific Comments: The authors demonstrate that H2K27M-mutant DMG and Ewing’s sarcoma tumor cells express high levels of GD2, which can be directly (via an anti-GD2 antibody) or indirectly (via inhibition of sphingolipid synthesis via eliglustat) targeted to potentially treat these tumor entities. However, there are several issues to be addressed in their study.
Title: The manuscript incorporates well the scope of the study.
Abstract: Overall, well-structured and concise. Comments:
- Simple Summary is nice but may be redundant with an Abstract. [minor]
- Statements on the action of the agents on proliferation or lipid synthesis are not supported by the data due to lack of statistical analyses. [major]
Introduction: This section presents background information on the GD2 and anti-GD2 antibody approaches. Comments:
- The authors present substantial background information on the GD2 but do not mention its specific relevance or importance to the tumor entities studied in the manuscript. [major]
- No background information on H2K27M-mutant DMG and Ewing’s sarcoma; what they are, current standard of treatment, need for new treatments. [major]
- Limited information on the therapeutic concept of indirect GD2 targeting by lipid synthesis inhibition. [minor]
Materials and Methods: The experimental procedures and study design are comprehensively explained. Comments:
- In section 4.9., the authors mention a graphical abstract but was not present anywhere in the manuscript files. [minor]
- There is no statistical analyses sections because such analyses were not performed. [major]
Results and Figures: The results are described well but the figures and legend require improvement. Comments:
- Some figures do not have figure legends or the figure legend are incorporated in the results text (See Figure 1, Figure 2, Figure 4). This authors should reformat this to avoid confusion. [major]
- In section 2.1., the authors mention “data not shown” for GM1 and GD1a expression. They should either remove any data refereeing to this, or preferably add the data in the supplement. I believe that all data should be shown in a manuscript. [major]
- In section 2.2, the authors mention GD2 expression assessment in U87-MG cells but miss citing Supplementary Figure 3, where these data is presented. [minor]
- In Figures 2,3, 4 and 6 (and wherever FACS data is shown), proper quantification of GD2 protein expression and statistical analyses should be performed. This can be done using software such as FCSExpress or FlowJO for FCAS quantification and PRISM graphad or SPSS for statistical analyses. It is also unclear to what is GD2 expression compered to and determined to be high or low? [major]
- Figure 4 refers to the GD2 expression in osteosarcoma cells, which are of no immediate interest to the study and therefore can be moved to the Supplement. This is also redundant since GD2 expression of osteosarcoma cells is outlined in Table 1. [minor]
- Section 2.4. is entitled “Mechanism of resistance to dinutuximab” but the authors do not present any in-depth mechanistical insight. Consider rephrasing this. [minor]
- Related to Figure 7 and Section 2.4., the authors should assess GD2 protein expression in the 400T cells (and compare it to that of 482T cells) to at least be able to speculate that a reduction in immune cell infiltrate correlated with a reduction in GD2 expression. [major]
- The reliability of results would significantly benefit fi the authors could at least show representative images of the immune cell staining used to calculate the numbers in Figure 7. Also, the authors should definitely perform statistical analyses to define whether this effect is significant and can support their claims. [major]
- Related to Figure 8, the authors should assess the GD2 protein expression in the eliglustat-treated cell to determine whether it does actually inhibit GD2. Otherwise, it could be any of the other lipids or steps outlined in Figure 8A. [major]
- Figure 8B is unclear. Consider reformatting or annotating it better and performing statistical la analyses. [major]
- There are 2 Sections 2.4. [minor]
Discussion and Conclusions: The authors discuss their findings with regards to published literature, but the length of this section could be significantly reduced to include additional discussion points. Comments:
- A potential limitation of this study is that cell culture conditions may indeed alter the expression of metabolic-related factors, such as GD2. The authors could discuss this and maybe why they did not measure GD2 expression in freshly isolated tumor samples. [minor]
- Disease control in the case study with a Ewing’s sarcoma patient was achieved in a combinatorial treatment including dinutuximab beta and, not in dinutuximab beta monotherapy. This should be pointed out more clearly in the discussion and the potential synergy with irinotecan/temozolomide discussed. [minor]
- The authors mention ST8SIA1 and B4GALNT1 out of context. They should provide an explanation as to what they are important (maybe reference Figure 8A?). [minor]
- The authors discuss Gaucher’s syndrome in too much detail. Since it is not even the scope of the study this section could be reduced. [minor]
- The authors show only descriptive (and not statistically significant) in vitro results on the effectiveness of eliglustat in inhibiting H2K27M-mutant DMG cell inhibition. They could propose or perform further pre-clinical in vivo studies to assess its efficacy in inhibiting tumor growth and then suggest moving to human patients. [minor]
Author Response
Abstract: Overall, well-structured and concise.
We thank the reviewer for his valuable feedback.
Comments:
Simple Summary is nice but may be redundant with an Abstract. [minor]
We have rewritten the simple summary as follow:
“Osteosarcoma, Ewing’s sarcoma and H3K27M-mutant diffuse midline glioma are rare but aggressive malignancies occurring mainly in children. Due to their rareness and often fatal course, drug development is challenging. Here, we repurposed the existing drugs dinutuximab and eliglustat and investigated their potential to directly target or indirectly modulate the tumor cell-specific ganglioside GD2. Our data suggest that targeting and/or modulating tumor cell-specific GD2 may offer a new therapeutic strategy for the above mentioned tumor entities”
Statements on the action of the agents on proliferation or lipid synthesis are not supported by the data due to lack of statistical analyses. [major]
We have added descriptive statistical analysis where it is suitable in the results (see below). However, please consider that most part of this work has been done in the framework of a personalized approach and therefore statistic cannot be always applied.
Introduction: This section presents background information on the GD2 and anti-GD2 antibody approaches.
Comments:
The authors present substantial background information on the GD2 but do not mention its specific relevance or importance to the tumor entities studied in the manuscript. [major]
We refined the introduction section in order to specify the relevance of GD2 in Ewing´ sarcoma and H3K27M-mutant midline glioma. We added the following (Line 92 ff.):
“In the pediatric population, GD2 has been discussed as target in rare and aggressive malignancies such as Osteosarcoma (OS), Ewing’s sarcoma (ES) and H3K27M-mutant diffuse midline glioma. ES represents the second common malignant bone tumor in children and young adults being associated with a poor outcome especially in patients with metastatic disease and even poorer in relapsed patients despite aggressive multimodal treatment regimens including surgery, chemotherapy and radiation [9]. In ES, GD2 has been suggested as target antigen to eradicate micrometastatic cells and prevent relapse in high-risk disease [10]. However, anti-GD2 antibody treatment is hardly used [11]. A phase I clinical trial (NCT00743496) from the St. Jude Clinical Research Hospital investigating the anti GD 2 antibody hu14.18K322A and aiming to enroll also ES patients was completed in 2014. However, recently published data considered neuroblastoma and osteosarcoma patients only [12]. Especially for H3K27M-mutant midline glioma, a rare malignant brain tumor in children and adults, novel treatment strategies are urgently needed. Radiotherapy is considered to be standard of care but provides even in combination with various systemic antineoplastic agents unsatisfactory treatment results with a median overall survival of around 12 month [13]. Recently, GD2 has been discussed as a target for CAR-T cell therapy in H3K27M-mutant DMG) [14]. In OS, GD2 has been suggested to play a role in chemotherapy resistance and tumor progression [15]. Phase I Clinical trials conducted with melanoma and osteosarcoma patients showed response to anti-GD2 antibody treatment but only in a fraction of the treated patients [16] [17]. This indicates the significance of a proper stratification to identify eligible patients for the anti-GD2 treatment.”
No background information on H2K27M-mutant DMG and Ewing’s sarcoma; what they are, current standard of treatment, need for new treatments. [major]
See previous point
Limited information on the therapeutic concept of indirect GD2 targeting by lipid synthesis inhibition. [minor]
We have added more background about indirect GD2 targeting (line 128 ff):
“Many studies indicate that tumor-associated gangliosides are a result of initial oncogenic transformation and play a key role in tumor progression [24]. Accordantly, inhibition of GD2 and GD3 synthesis has been suggested as a therapeutic approach in several cancers [25] [26]. However, methods to inhibit gangliosides in a clinical setting in the context of cancer are largely missing”
Materials and Methods: The experimental procedures and study design are comprehensively explained.
Comments:
In section 4.9., the authors mention a graphical abstract but was not present anywhere in the manuscript files. [minor]
The graphical abstract was included in the original word document. We have no explanation why it was not accessible to the reviewer. We have included it again in the revised manuscript at the end of the document and also send it as a Figure.
There is no statistical analyses sections because such analyses were not performed. [major]
We have added descriptive statistical analysis where it is suitable in the results.
Results and Figures: The results are described well but the figures and legend require improvement.
Comments:
Some figures do not have figure legends or the figure legend are incorporated in the results text (See Figure 1, Figure 2, Figure 4). This authors should reformat this to avoid confusion. [major]
In the revised manuscript figure legends have been reformatted.
In section 2.1., the authors mention “data not shown” for GM1 and GD1a expression. They should either remove any data refereeing to this, or preferably add the data in the supplement. I believe that all data should be shown in a manuscript. [major]
We have added the data in Supplemental Figure 1
In section 2.2, the authors mention GD2 expression assessment in U87-MG cells but miss citing Supplementary Figure 3, where these data is presented. [minor]
We apologize for the error and have corrected it
In Figures 2,3, 4 and 6 (and wherever FACS data is shown), proper quantification of GD2 protein expression and statistical analyses should be performed. This can be done using software such as FCSExpress or FlowJO for FCAS quantification and PRISM graphad or SPSS for statistical analyses. It is also unclear to what is GD2 expression compered to and determined to be high or low? [major]
The FACS data analysis was performed using FlowJo as described in the methods section. Indeed, we do not show further statistical analysis. We describe a huge variation of GD2 expression in the different samples, especially in osteosarcoma. One of the aims of the presented results is to show the heterogeneity for GD2 expression within a tumor entity with similar histological appearance. We now make this clear by highlighting the need of “proper stratification to identify eligible patients for the anti-GD2 treatment” in the introduction section. In addition, we provide the percentage of GD2 positive population to highlight the heterogenic expression within the same tumor cell population. According data are now available in table 1. The gating strategy is shown in supplemental figure 2. We used the percent of GD2 positive cells to score GD2 expression as shown in the legend of Table 1.
Figure 4 refers to the GD2 expression in osteosarcoma cells, which are of no immediate interest to the study and therefore can be moved to the Supplement. This is also redundant since GD2 expression of osteosarcoma cells is outlined in Table 1. [minor]
We think that visualization of the flow cytometry data is important to understand the heterogeneity in the expression of GD2. Therefore, we left this Figure in the manuscript and not in the supplemental
Section 2.4. is entitled “Mechanism of resistance to dinutuximab” but the authors do not present any in-depth mechanistical insight. Consider rephrasing this. [minor]
We don’t show indeed mechanistical insight. Therefore, we have changed the title to:
“GD2 expression and immune cells infiltration in the dinutuximab-refractory metastasis”
Related to Figure 7 and Section 2.4., the authors should assess GD2 protein expression in the 400T cells (and compare it to that of 482T cells) to at least be able to speculate that a reduction in immune cell infiltrate correlated with a reduction in GD2 expression. [major]
For sample 400T only FFPE material was available. As discussed in the discussion (line 415 ff), the specificity of GD2 detection on FFPE tissues by immunohistochemistry is controversial and therefore information on GD2 expression cannot be shown in this sample. Moreover, our aim was to compare a tumor without engagement of antibody effector cells with a tumor under treatment with a monoclonal antibody which should engage immune cells expressing a Fc receptor. We have also discussed in more details the immune cells infiltrate in the discussion (line 521 ff).
The reliability of results would significantly benefit fi the authors could at least show representative images of the immune cell staining used to calculate the numbers in Figure 7. Also, the authors should definitely perform statistical analyses to define whether this effect is significant and can support their claims. [major]
We have included representative images in supplemental Figure 9. Because we analysed only one patient, we can not conclude if an immune cell population associates with the efficacy of the therapy. We can only describe what we observed in our case. Moreover, heterogeneity within a tumor has also to be considered. To describe this heterogeneity, we we have included the min/max with mean for two tumor regions per time point. And adapted the results part as:
“We analyzed the immune cell infiltrates in longitudinal samples (400T versus 482T), and distinguished between tumor and non-tumor (i.e. stroma and necrosis, Supplemental figure 8). Immune cells infiltration was heterogeneous within the same tumor, but we observed a tendency to decreased numbers of B cells, NK cells and macrophages in the sample that was resected after dinutuximab treatment (Figure 8 ).
Related to Figure 8, the authors should assess the GD2 protein expression in the eliglustat-treated cell to determine whether it does actually inhibit GD2. Otherwise, it could be any of the other lipids or steps outlined in Figure 8A. [major]
This is indeed an interesting experiment, but unfortunately not feasible in the 5 revision days. We are planning to investigate this subject in further experiments. Eliglustat has been shown to alter the glycosphingolipid composition as discussed at line 550. Moreover, we think that GD2 expression may be an indicator of a generally deregulated gangliosides synthesis. Indeed we discussed that: “Here we hypothesized that due to the extremely high expression of GD2 in H3K27M-mutant DMG, the glyco-sphingolipids metabolism is particularly active in these cells and that its perturbation via inhibition of the glucosylceramide synthase could be lethal” at line 559 ff
Figure 8B is unclear. Consider reformatting or annotating it better and performing statistical la analyses. [major].
We reformatted this figure and added descriptive statistics by showing the mean IC50 values and the range for eliglustat treated tumor cells. Each patient derived tumor cell line was treated twice for validation. As mentioned before, most part of this work has been done in the framework of a personalized approach. In our opinion it could be misleading to perform statistical analysis for a group of two. Since we show dater for a tumor entity which is very rare, this is still important.
There are 2 Sections 2.4. [minor]
We have corrected this error
Discussion and Conclusions: The authors discuss their findings with regards to published literature, but the length of this section could be significantly reduced to include additional discussion points.
Comments:
A potential limitation of this study is that cell culture conditions may indeed alter the expression of metabolic-related factors, such as GD2. The authors could discuss this and maybe why they did not measure GD2 expression in freshly isolated tumor samples. [minor]
This is indeed an interesting point. However, we would expect that the culture conditions induce a similar expression within the same tumor type while with exception of DIPG, GD2 expression was rather heterogeneous within the same entity. To address expression AND heterogeneity of GD2 expression in a tumor sample, immunohistochemistry would be necessary. However, gangliosides are soluble in some solvents used to fix tissues and GD2 expression cannot be assessed by IHC on FFPE samples. An alternative would be to use FF samples. We have discussed this as (Line 424):
„Detection of GD2 by IHC on frozen tumor samples could also represent a valid alternative as the tumor is not manipulated before GD2 analysis [41].
Disease control in the case study with a Ewing’s sarcoma patient was achieved in a combinatorial treatment including dinutuximab beta and, not in dinutuximab beta monotherapy. This should be pointed out more clearly in the discussion and the potential synergy with irinotecan/temozolomide discussed. [minor]
According to the reviewer´s recommendation, we pointed out the impact of the combination treatment of dinutuximab beta and IT more thoroughly in the discussion section. Additionally, the potential synergistic effect was discussed.
We added the following:
“We applied dinutuximab beta in combination with IT to a patient with refractory ES whose tumor had a high and homogenous expression of GD2.” (line 446)
“Additionally, there is strong evidence that various cytotoxic agents are able to stimulate anticancer immune responses involving NK cells and CD8 positive T cells and thus to augment immunotherapeutic effects.” (line 454)
“Given this poor outcome, achieved treatment result in our patient by incorporating dinutuximab beta in an individualized treatment approach is quite remarkable indicating a synergistic antineoplastic effect of IT and anti GD 2 antibodies in GD 2 expressing ES.” (line 465 ff)
“This experience further supports the need for a prospective clinical evaluation of anti GD 2 antibodies in combination with a sufficient conventional cytotoxic treatment in advanced ES, like recently recommended by the COG task force.” (line 474 ff)
The authors mention ST8SIA1 and B4GALNT1 out of context. They should provide an explanation as to what they are important (maybe reference Figure 8A?). [minor]
We have changed the sentence as following: „As we observed an increase in GM3 expression in the metastasis, reduction in GD2 expression might be associated with a down regulation of ST8SIA1 and B4GALNT1, which are required to transform available GD3 into GT3 or GM2 respectively (Figure 1C) Recently a 2-gene signature composed of ST8SIA1 + B4GALNT1 has been suggested as efficient predictor of GD2-positive phenotype [53].“
The authors discuss Gaucher’s syndrome in too much detail. Since it is not even the scope of the study this section could be reduced. [minor]
We have rewritten this section, also to address questions of reviewer 2.
The authors show only descriptive (and not statistically significant) in vitro results on the effectiveness of eliglustat in inhibiting H2K27M-mutant DMG cell inhibition. They could propose or perform further pre-clinical in vivo studies to assess its efficacy in inhibiting tumor growth and then suggest moving to human patients. [minor]
We thank the reviewer for the suggestion. In the conclusions we have added:
„Further pre-clinical in vivo analyses are required to elucidate the mechanism of action of glucosylceramide synthase inhibitors on tumor growth and will facilitate the access of pediatric patients to innovative clinical studies“
Round 2
Reviewer 3 Report
The authors manage to address the majority of the reviewer’s comments and the manuscript has improved significantly. The manuscript provides novel insights and would be of interest to readers. However, major concerns remain unaddressed with no adequate justification by the authors. See comments below:
The authors fail to provide any statistical analyses of their data (where applicable) to support their conclusions. It is unclear to me, what the phrase “most part of this work was done in the framework of a personalized approach and therefore statistic cannot be always applied”. Although, I understand that for some parts of the study limited patient samples were available and analyzed, it does not justify the complete lack of inferential statistical analyses:
- The heterogeneity of GD2 expression between tumor samples is clear but can be further supported by actually comparing it between samples. In Figures 3, 4 and 5, where samples were analyzed by FACS for the protein expression of GD2, the authors should perform statistical analyses (a simple t-test or one-way ANOVA) to compare and determine whether GD2 expression significantly differs between patient samples. The FACS data provides a cumulative readout of 2x105 – 5x105 cells (See section 4.3) per sample and should suffice for inferential statistical analyses to provide a statistical p-value for the comparisons. See PMID:30414924, Fig. 3K as example, where the authors compared the % of Annexin positive cells between samples. For example, in Figure 7, similar methodology can be applied to statistically compare the % expression of the different ganglioside entities.
- In Figure 8, the authors provide a mean and min/max for each immune cell population quantification but NOT the corresponding statistical test to compare between Tumor and Non-tumor and/or 400T and 482T. Again, a simple t-test (or non-parametric analogous test) should suffice for an inferential statistical comparison between two groups.
- In Figure 9, the authors claim that the experiments to determine the IC50 for each treatment were done only twice. Why didn’t the authors repeat the experiment at least three times to facilitate proper statistical analyses? The presented data, of only two biological replicates, may then not be convincing. If that is the case, the authors should clearly mention that in their manuscript to avoid misleading the readership or perform an additional in vitro experiment and provide inferential statistical analyses for the treatment comparisons. This would also depend on the flexibility of the journal to provide additional (more than five days) to adequately revise the manuscript.
Author Response
- The heterogeneity of GD2 expression between tumor samples is clear but can be further supported by actually comparing it between samples. In Figures 3, 4 and 5, where samples were analyzed by FACS for the protein expression of GD2, the authors should perform statistical analyses (a simple t-test or one-way ANOVA) to compare and determine whether GD2 expression significantly differs between patient samples. The FACS data provides a cumulative readout of 2x105 – 5x105 cells (See section 4.3) per sample and should suffice for inferential statistical analyses to provide a statistical p-value for the comparisons. See PMID:30414924, Fig. 3K as example, where the authors compared the % of Annexin positive cells between samples.
We think that the comparison of GD2 expression within the same tumor entity does not need a statistical analysis because we are neither comparing two conditions (for example stimulated vs not stimulated) nor subtypes within an entity. The difference between entities may be interesting for the readership. We used one way ANOVA and Tukey’s multiple comparison test to compare the different entities. A statistically significant difference between the groups was found (p=0.0199) and the difference between OS and H3K27-mutant DMG had an adjusted p value of 0.00384. However, the power of the analysis was low due to the small samples size. We added this result at line 226.
- For example, in Figure 7, similar methodology can be applied to statistically compare the % expression of the different ganglioside entities.
In Figure 7, a tumor sample (either 408T or 482T) was used to extract lipids and analyse the gangliosides composition. Unfortunately, it is not possible to perform statistic with two groups where each group contains only one sample. Due to the small size of fresh frozen tissues available after surgery, it was not possible to analyze more than one aliquot. We have pointed this out at line 347 (legend Figure 7)
- In Figure 8, the authors provide a mean and min/max for each immune cell population quantification but NOT the corresponding statistical test to compare between Tumor and Non-tumor and/or 400T and 482T. Again, a simple t-test (or non-parametric analogous test) should suffice for an inferential statistical comparison between two groups.
According to our understanding, it is not possible to perform inferential statistics for a clinical case study with longitudinal data (before and after treatment) of one patient. We assessed leukocyte densities in two different areas per time point to consider infiltration heterogeneity, however, these were technical replicates, which are not statistically independent and have to be summarized to avoid pseudoreplication (Lazic BMC Neursci. 2010; PMID: 20074371). We presented a study of a single person (aka N-of-1 trial) which is an additional accepted way to assess individual treatment responses (Schork Nature 2015; PMID: 25925459), particularly in rare clinical settings. N-of-1 trials typically report graphical data to allow for a visual inspection of effect sizes (Lobo et al. J Neurol Phys 2017; PMID: 28628553). The generalizability of the observed effects could be achieved later by combining effect sizes across more cases. If the reviewer is asking for statistically significant difference within a sample (for example tumor vs non-tumor in the same sample) a t-test may be suitable. However, because only 2 values/group were available a statistical analysis would not be exact, due to the impossibility to assess the distribution (parametric vs non-parametric test).
- In Figure 9, the authors claim that the experiments to determine the IC50 for each treatment were done only twice. Why didn’t the authors repeat the experiment at least three times to facilitate proper statistical analyses? The presented data, of only two biological replicates, may then not be convincing. If that is the case, the authors should clearly mention that in their manuscript to avoid misleading the readership or perform an additional in vitro experiment and provide inferential statistical analyses for the treatment comparisons. This would also depend on the flexibility of the journal to provide additional (more than five days) to adequately revise the manuscript.
- H3K27M-mutant DMG is anatomically located in the brainstem in the pons region. Performing surgery in this tumor entity is impossible due to the direct proximity to vital essential CNS regions (e.g. cardio regulatory and respiratory centers). Therefore, samples were isolated from one needle biopsy, which is only performed by experienced pediatric neurosurgeons. Therefore, the number of cells that could be used for experiments is limited. We managed to perform characterization of the primary cells and proliferation (in triplicates for each condition) of the primary cells, but unfortunately, we haven’t had enough cells to repeat the experiment three times. This is of course a limitation when working with primary cells at low passage. We corrected the legend at line 392 in the legend of Figure 9 and 734 in material and methods to avoid misleading the readership. Moreover, we added in Material and Methods at line 605 that the cells were isolated from a needle biopsy.
- We have added a statistical analysis section at line 780 in material and methods.
Difference in GD2 expression between tumor entities was calculated with one way ANOVA and Tukey’s multiple comparison test. GraphPad Prism was used to perform the analysis.